# Safety, immunogenicity and efficacy of the self-amplifying mRNA ARCT-154 COVID-19 vaccine: pooled phase 1, 2, 3a and 3b randomized, controlled trials

Combination of waning immunity and lower effectiveness against new SARS-CoV-2 variants of approved COVID-19 vaccines necessitates new vaccines. We evaluated two doses, 28 days apart, of ARCT-154, a self-amplifying mRNA COVID-19 vaccine, compared with saline placebo in an integrated phase 1/2/3a/3b controlled, observer-blind trial in Vietnamese adults (ClinicalTrial.gov identifier: NCT05012943). Primary safety and reactogenicity outcomes were unsolicited adverse events (AE) 28 days after each dose, solicited local and systemic AE 7 days after each dose, and serious AEs throughout the study. Primary immunogenicity outcome was the immune response as neutralizing antibodies 28 days after the second dose. Efficacy against COVID-19 was assessed as primary and secondary outcomes in phase 3b. ARCT-154 was well tolerated with generally mild–moderate transient AEs. Four weeks after the second dose 94.1% (95% CI: 92.1–95.8) of vaccinees seroconverted for neutralizing antibodies, with a geometric mean-fold rise from baseline of 14.5 (95% CI: 13.6–15.5). Of 640 cases of confirmed COVID-19 eligible for efficacy analysis most were due to the Delta (B.1.617.2) variant. Efficacy of ARCT-154 was 56.6% (95% CI: 48.7– 63.3) against any COVID-19, and 95.3% (80.5–98.9) against severe COVID-19. ARCT-154 vaccination is well tolerated, immunogenic and efficacious, particularly against severe COVID-19 disease.

The global COVID-19 pandemic disease is now largely under control and no longer considered a public health emergency of international concern (PHEIC)[1]. The unprecedented rapid development and distribution of several highly effective vaccines against the severe acute respiratory syndrome coronavirus 2 virus (SARS-CoV-2) has limited the advance of the global pandemic. However, a major public health burden remains due to outbreaks of COVID-19 caused by new SARS-CoV-2 Variants of Concern[2]. Accumulation of mutations in the spike glycoprotein (S protein), the main antigenic target, has made each successive new variant less susceptible to vaccine-induced immunity[3], which combined with waning vaccine immunity has

contributed to declining effectiveness of the currently available vaccines against new variants[4–8]. Vaccine effectiveness against the ancestral SARS-CoV-2 of two doses of either of the two main licensed mRNA vaccines, BNT162b2 and mRNA-1273, has been calculated to decline to 50.8% (95% CI: 19.7, 69.8) by 7 months after the second dose[9]. Further, the most recently emerged Omicron variants cause less severe disease but are more infectious and easily transmissible[10,11], leading to concerns that future variants may combine their high transmissibility with the severe disease of the original virus. These concerns are driving development of new vaccines that can elicit both greater breadth against new variants and longer

e-mail: v.hungnx1@vinmec.com

duration of humoral immunity before antibodies wane to maintain protection against future COVID outbreaks.

One technology being applied in this development is the use of self-amplifying mRNA (sa-mRNA) that allows host cells to make copies of the vaccine mRNA, increasing the amount of protein produced with lower doses of administered mRNA[12–15]. Preclinical studies have shown sa-mRNA vaccines elicit a durable and broader activation of the immune response[16]. sa-mRNA COVID-19 vaccines were safe and immunogenic in human phase 1 studies – a dose-ranging primary vaccination study[17] and a small booster study in older adults[18]. Arcturus Therapeutics, Inc (Arcturus, San Diego, CA, USA) has developed a group of sa-mRNA COVID-19 vaccines, including ARCT-154, which encode the SARS-CoV-2 S glycoprotein[19]. Preclinical toxicology studies and an interrupted phase 2 clinical study (ClinicalTrials.gov identifier: NCT04668339) of the predecessor vaccine, ARCT-021, which encodes the Wuhan-Hu-1 strain S-protein, showed it was safe. In three phase 1/2 clinical trials involving >500 treated adults ARCT-021 had acceptable safety and tolerability profiles and two doses of 5 µg or more were shown to be immunogenic when administered 4 weeks apart[19]. ARCT-154 was then developed based on the S-protein with the D614G mutation, a proline substitution resulting in the S-protein being expressed in the prefusion conformation and furin cleavage site modification to improve stability. Additional changes include optimization of the replicon and modification of the vaccine impurity profile which were associated with increased immunogenicity and an improved tolerability profile compared with the parent vaccine in preclinical studies. Based on the human clinical experience with ARCT-021 we initiated the present accelerated, integrated phase 1/2/3a/3b study, designed following EMA, FDA and WHO guidance, to evaluate the safety, reactogenicity, immunogenicity, and efficacy of ARCT-154. We present the first study results up to three months after the first vaccination of human volunteers with this vaccine.

## Results

### Study participants
This randomized, double-blind, controlled phase 1, 2, 3a, and 3b integrated study is ongoing at 16 study centers in Vietnam (Supplementary Table 1) where all study participants are being monitored until one year after their second vaccination; those results will be reported separately. This report covers data obtained from enrolment of the first phase 1 participant on 15 August 2021 until 12 January 2023, the cut-off for data extraction for interim safety and per protocol efficacy analyzes from phase 3b. Basic details of the four phases integrated into the whole study are illustrated in Fig. 1. Enrollment was sequential from phase 1, approval for enrollment into subsequent phases only being given following analysis of initial safety data by an independent data safety monitoring board (DSMB). Demographics of participants in each study phase were generally similar across study groups in terms of gender, weight and BMI (Table 1). Median age was lowest in phase 1 in which only 18–60 years olds were enrolled, and highest in the phase 3b efficacy study as those ≥ 60 years of age were included. Study compliance was excellent across all phases, particularly in phases 1, 2, and 3a in which 99 of 100 (99%), 300 of 301 (99.7%), and 579 of 600 (96.5%) received their second dose, respectively (Table 1). In phase 3b, 7869 of 8056 (97.6%) received their second dose of ARCT-154 compared with 7831 of 8044 (97.3%) placebo recipients. At Day 1 in phases 1, 2, and 3a all participants were negative for anti-nucleocapsid antibody while in phase 3b, 99.4% of participants were negative (Table 1).

### Safety and reactogenicity
Across phases 1, 2, and 3a 1001 participants received at least one dose of their allocated study treatment; of these 670 of 748 (89.6%) ARCT-154 vaccinees and 136 of 253 (53.8%) placebo recipients reported at least one adverse event after the first dose (Table 2). These rates declined slightly after the second dose but remained higher in

vaccinees than placebo recipients. The majority of reported adverse events were solicited local reactions, mainly mild or moderate injection site pain or tenderness with few reports of swelling, induration or erythema (Fig. 3). Most local reactions occurred within three days of vaccination and resolved within 2-4 days after either dose (Supplementary Fig. 1). Solicited systemic adverse events were mainly mild or moderate in severity and also occurred within days 1–3 post-dose 1 or 2 and resolved within the follow-up period. Rates of most solicited systemic adverse events were higher in vaccinees than placebo recipients, the most frequent being fatigue, myalgia, headache, arthralgia and chills; unlike local reactions, rates of systemic adverse events were not markedly lower after the second dose compared with the first dose for both groups. In the larger phase 3b, rates of solicited AEs were higher in vaccinees than placebo recipients but lower than in phases 1, 2, and 3a (Fig. 4), and declined slightly after the second dose. For all doses AEs were mainly composed of mild to moderate local pain and tenderness, headache, fatigue and myalgia.

Incidences of unsolicited adverse events up to 28 days after each dose were similar in vaccinees and placebo recipients (Table 2). In phases 1, 2, and 3a rates were lower after the second dose than the first, but in phase 3b rates were similar after doses 1 and 2. Incidence rates of AEs considered related to study injections were all below 5% and rates of severe AEs were generally below 1% and similar in vaccine and placebo groups.

There were 30 serious adverse events reported in phases 1, 2, and 3a combined; 14 in 748 (1.9%) vaccinees and 16 in 253 (6.3%) placebo recipients; only two, both in placebo recipients, were considered to be related to study injections and led to discontinuation from the study. In the larger phase 3b study there were 319 serious adverse events, 118 in 8059 (1.5%) vaccinees and 201 in 8041 (2.5%) placebo controls. Fifteen serious adverse events were related to study injections, 10 (0.1%) to vaccine and 5 (0.1%) to placebo. There were no deaths in phases 1, 2, and 3a, but 21 deaths occurred in phase 3b, of 5 vaccinees and 16 placebo recipients. Of these, none were related to vaccination but 10 were considered to be associated with COVID-19 infection, one in a vaccinee and nine in placebo recipients.

### Immunogenicity
Baseline immune responses assessed in 965 participants in phases 1, 2, and 3a, showed similar levels of surrogate virus neutralization antibody titers (sVNT) in ARCT-154 (n = 723) and placebo (n = 242) groups before vaccination (Fig. 5). Geometric mean concentrations (GMCs) in placebo recipients did not change by Days 29 or 56; 4 had seroconverted by Day 29 and 1 by Day 57. In contrast, ARCT-154 was highly immunogenic, with 386 of 717 (53.8%) vaccinees seroconverting by Day 29, four weeks after the first dose, and 658 of 699 (94.1%) seroconverting by Day 57, four weeks after the second dose. sVNT GMCs increased at each time point, with geometric mean-fold rises (GMFR) of 4.0 (95% CI: 3.7–4.2) at Day 29 and 14.5 (13.6–15.5) at Day 57. These sVNT immune responses were confirmed by the validated D614G microneutralization assay. This showed 375 of 391 (95.9%) vaccinees seroconverted by Day 57 after two doses of ARCT-154 compared with 3 of 131 (2.3%) placebo recipients (Fig. 5). The GMFR was 20.9 (95% CI: 19.2–22.9) in the ARCT-154 group and 1.2 (1.1–1.3) in the placebo group.

### Vaccine efficacy (VE)
In Phase 3b there were 3632 (1652 ARCT-154, 1980 placebo) suspected cases of COVID-19 reported from Day 1 to Day 92. Nasopharyngeal swabs were collected from 1920/3632 (52.9%) cases within 3 days, 978/3632 (26.9%) within 4–7 days and 438/3632 (12.1%) within 8–14 days of symptom onset. An independent expert Event Adjudication Committee (EAC) adjudicated 836 virologically-confirmed cases and assessed 734 cases as COVID-19 disease (including 48 cases of severe COVID-19 and 10 deaths attributed to COVID-19), and 102 cases as asymptomatic SARS-CoV-2 infection. Of these confirmed COVID-19 cases, 643

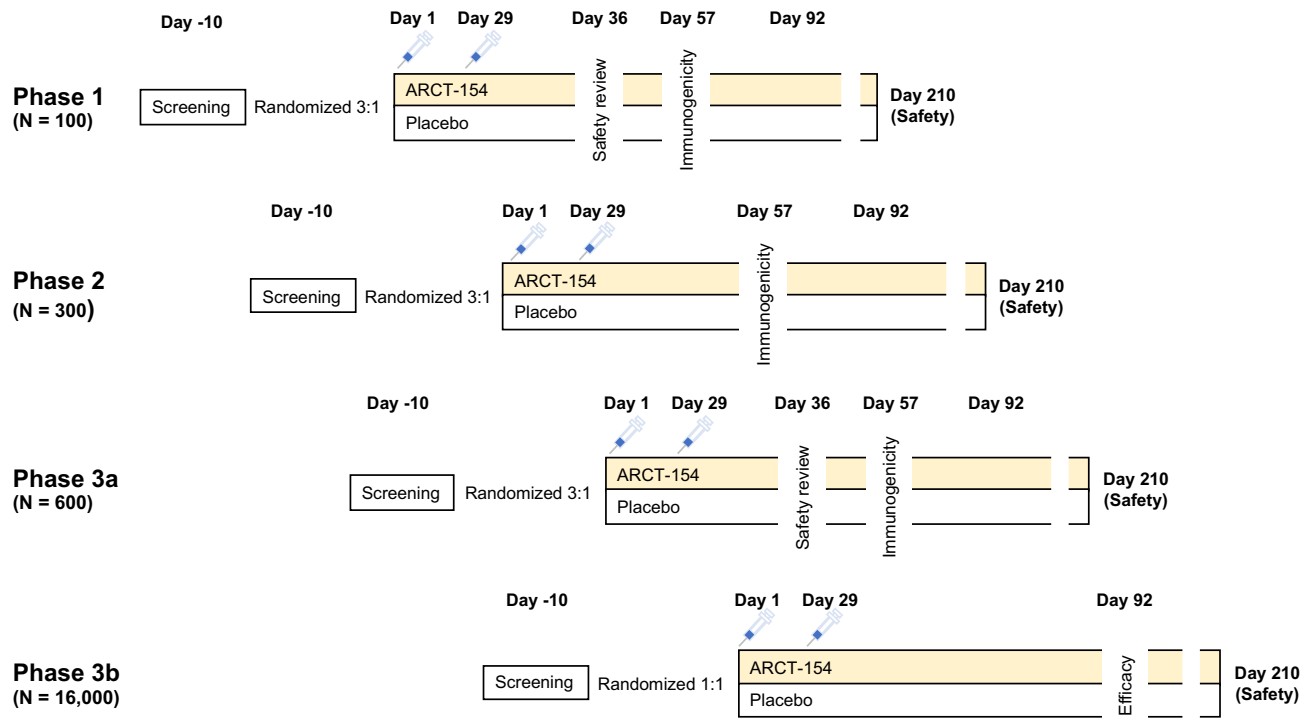

**Fig. 1 | Study design.** The different phases (1, 2, 3a and 3b) of the study are shown with randomization of ARCT-154 and placebo groups, and timings of interventions.

**Table 1 | Baseline characteristics of study participants enrolled in the different phases (ITT, as Randomized)**

| | | | Phase 1 | | Phase 2 | | Phase 3a | | Phase 3b | |
|---|---|---|---|---|---|---|---|---|---|---|
| | | | ARCT-154 (*N* = 75) | Placebo (*N* = 25) | ARCT-154 (*N* = 226) | Placebo (*N* = 75) | ARCT-154 (*N* = 448) | Placebo (*N* = 152) | ARCT-154 (*N* = 8059) [a] | Placebo (*N* = 8048) [a] |
| **Female gender** | | *n* (%) | 35 (46.7) | 14 (56.0) | 109 (48.2) | 36 (48.0) | 194 (43.3) | 67 (44.1) | 4099 (50.9) | 4088 (50.8) |
| **Age** (years) | | Median (Q1, Q3) | 31 (25, 40) | 33 (27, 43) | 38 (28, 49) | 42 (29, 53) | 45 (36, 54) | 43 (34, 55) | 48 (37, 57) | 48 (37, 57) |
| **Ethnicity** | Kinh | *n* (%) | 74 (98.7) | 24 (96.0) | 217 (96.0) | 75 (100) | 443 (98.9) | 148 (97.4) | 8025 (99.6) | 8001 (99.4) |
| | Others | | 1 (1.3) | 1 (4.0) | 9 (4.0) | 0 (0) | 5 (1.1) | 4 (2.6) | 34 (0.4) | 47 (0.6) |
| **Weight** (kg) | | Median (Q1, Q3) | 59 (50, 68) | 55 (49, 62) | 58 (51, 66) | 58 (50, 64) | 60 (53, 67) | 58 (50, 67) | 56 (51, 63) | 56 (50, 63) |
| **BMI** (kg/cm²) | | Median (Q1, Q3) | 23 (19, 25) | 22 (19, 24) | 23 (21, 26) | 23 (21, 25) | 24 (22, 26) | 23 (21, 26) | 23 (21, 25) | 23 (21, 25) |
| **Risk of severe COVID-19** [b] | | | | | | | | | | |
| 18-59- year-olds with co-morbidities | | *n* (%) | 0 (0) | 0 (0) | 70 (31.0) | 25 (33.3) | 177 (39.5) | 59 (38.8) | 2816 (34.9) | 2810 (34.9) |
| ≥ 60 years | | | 0 (0) | 0 (0) | 22 (9.7) | 8 (10.7) | 58 (12.9) | 21 (13.8) | 1401 (17.4) | 1401 (17.4) |
| **Vaccination compliance** | Received dose 1 | *n* (%) | 75 (100) | 25 (100) | 226 (100) | 75 (100) | 448 (100) | 152 (100) | 8056 (100) | 8044 (100) |
| | Received dose 2 | | 74 (98.7) | 25 (100) | 225 (99.6) | 75 (100) | 434 (96.9) | 145 (95.4) | 7869 (97.6) | 7831 (97.3) |
| **Anti-N protein antibody negative** | | *n* (%) | In phases 1, 2 and 3a all (100%) ARCT-154 participants & all (100%) Placebo participants were negative | | | | | | 8005 (99.3) | 7999 (99.4) |
| **Anti-N protein antibody positive** | | *n* (%) | | | | | | | 54 (0.7) | 49 (0.6) |

[a]Of 8048 participants randomized to receive placebo 8 received vaccine, and 1 randomized to vaccine received placebo so 8041 effectively received placebo.
[b]Risk based on presence of predefined medical conditions known to be associated with risk of severe COVID-19.

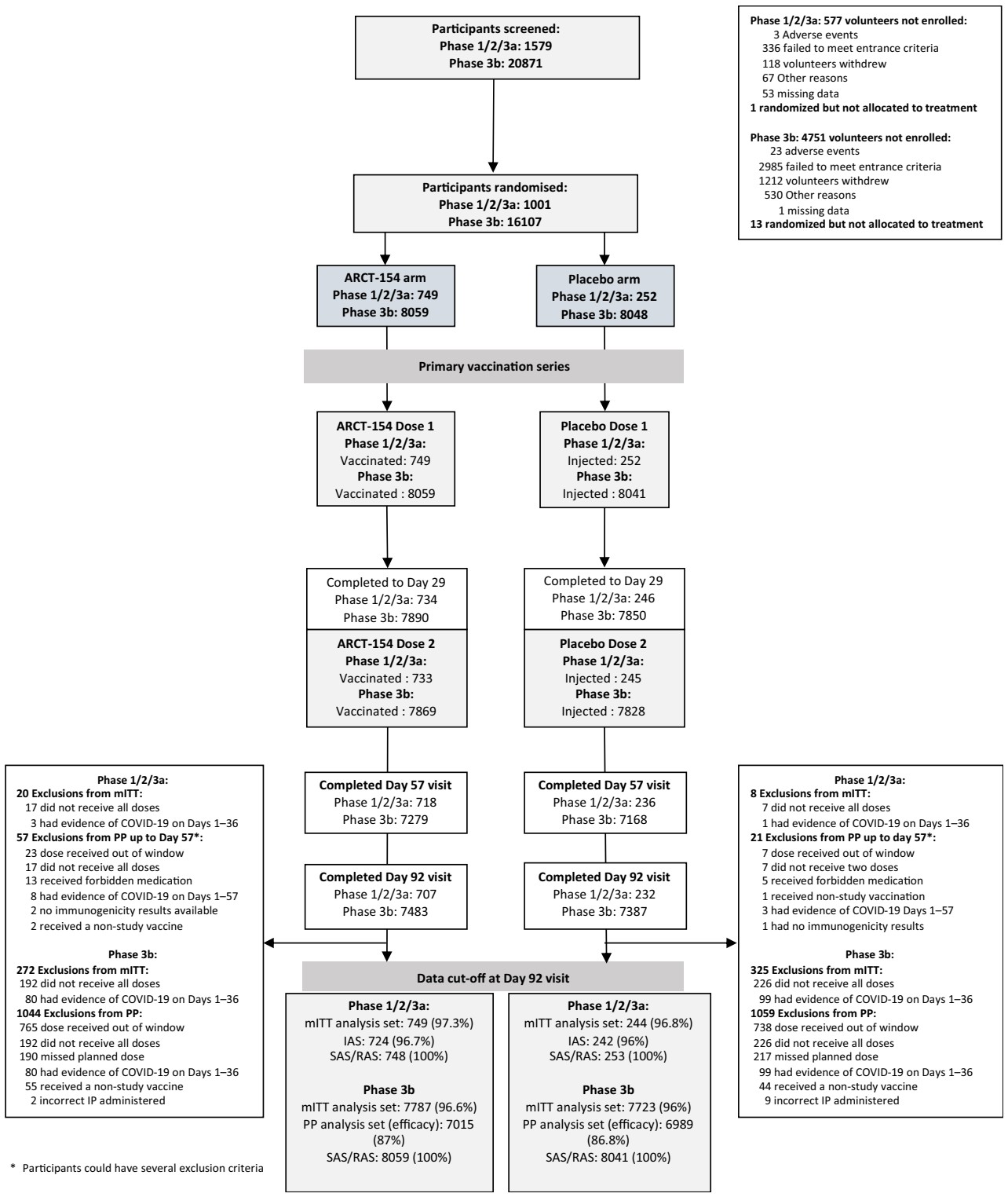

**Fig. 2 | Study flow chart showing the disposition of participants throughout the various phases of the study.** Numbers and percentages are based on those randomized; participants could be excluded from the analysis sets for more than one reason. Analysis sets: ITT, Intent-to-treat; mITT, modified Intent-to-treat; PP, Per protocol set; RAS, Reactogenicity analysis set; SAS, Safety analysis set; IAS, immunogenicity analysis set. In Phase 1/2/3a 1 participant randomized to ARCT-154 received Placebo as Dose 1 giving 748 receiving ARCT-154 and 253 receiving Placebo as Dose 1. Phase 3b: 7 participants randomized to Placebo did not receive Dose 1 giving 8041 in the SAS Placebo group; two participants who received ARCT-154 as Dose 1 but placebo as Dose 2 were removed from the ARCT-154 Dose 2 SAS arm giving 7867, and 6 participants randomized to Placebo received incorrect investigational product and were removed from the Placebo Dose 2 SAS arm giving 7822.

occurred from Day 36 to Day 92, but in three cases participants had received a non-study COVID-19 vaccine so were excluded from the modified Intention to Treat (mITT) set for analysis, leaving 640 eligible mITT cases (200 vaccinees and 440 placebo recipients), including 43 severe cases and 10 deaths attributed to COVID-19 (Table 3, Fig. 6). The primary vaccine efficacy (VE) objective was met as two doses of ARCT-154 had an efficacy of 56.6% (95% CI: 48.7–63.3) against COVID-19 disease of any severity with the lower bound of the CI above the

**Table 2 | Adverse events after doses 1 and 2 of ARCT-154 and placebo in phases 1, 2 and 3a combined, and phase 3b (Safety Set, as treated)**

| N = first dose / second dose | | Phases 1, 2 and 3a | | Phase 3b | |
|---|---|---|---|---|---|
| | | ARCT-154 (N = 748[a] / 732) | Placebo (N = 253[a] / 245) | ARCT-154 (N = 8059 / 7867[c]) | Placebo (N = 8041[b] / 7822[c]) |
| **Any solicited adverse event,[d]** n (%) | Dose 1 | 670 (89.6) | 136 (53.8) | 4732 (59.7) | 2768 (35.1) |
| | Dose 2 | 582 (79.5) | 104 (42.4) | 3833 (49.8) | 2006 (26.3) |
| Local reactions, n (%) | Dose 1 | 586 (78.3) | 51 (20.2) | 3474 (43.8) | 858 (10.9) |
| | Dose 2 | 452 (61.7) | 28 (11.4) | 2401 (31.2) | 585 (7.7) |
| Systemic adverse events, n (%) | Dose 1 | 557 (74.5) | 120 (47.4) | 3816 (48.1) | 2499 (31.7) |
| | Dose 2 | 506 (69.1) | 93 (38.0) | 3214 (41.7) | 1796 (23.5) |
| **Any adverse event within 28 days[e]** | Dose 1 | 177 (23.7) | 71 (28.1) | 1125 (14.0) | 1101 (13.7) |
| | Dose 2 | 124 (16.9) | 45 (18.4) | 1096 (13.9) | 1241 (15.9) |
| Related adverse event within 28 days, n (%) | Dose 1 | 27 (3.6) | 11 (4.3) | 202 (2.5) | 184 (2.3) |
| | Dose 2 | 19 (2.6) | 5 (2.0) | 130 (1.7) | 107 (1.4) |
| Severe adverse event within 28 days, n (%) | Dose 1 | 1 (0.1) | 0 | 10 (0.1) | 18 (0.2) |
| | Dose 2 | 0 | 0 | 13 (0.2) | 17 (0.2) |
| **Serious adverse event (SAE) to switch-over[f]** n (%) | | 14 (1.9) | 16 (6.3) | 118 (1.5) | 201 (2.5) |
| Related serious adverse event | | 0 | 2 (0.8) | 10 (0.1) | 5 (0.1) |
| SAE leading to discontinuation | | 0 | 2 (0.8) | 8 (0.1) | 15 (0.2) |
| **Medically-attended adverse event to switch-over[f]** n (%) | | 114 (15.2) | 57 (22.5) | 975 (12.1) | 1178 (14.6) |
| Related medically-attended adverse event | | 5 (0.7) | 4 (1.6) | 91 (1.1) | 63 (0.8) |
| **Death** n (%) | | 0 | 0 | 5 (0.1) | 16 (0.2) |

[a]In Phase 1/2/3a 1 participant randomized to ARCT-154 who erroneously received placebo was included in the placebo group for safety analysis.

[b]In Phase 3b, 7 "placebo" participants did not receive placebo and were excluded from safety analysis.

[c]Eight participants (2 ARCT-154 and 6 placebo) received incorrect study product as Dose 2 and were excluded from the safety analysis post-Dose 2.

[d]Solicited adverse events occurring within 7 days of vaccination.

[e]Adverse events reported within 28 days of each vaccination.

[f]Serious and medically-attended adverse events recorded from Day 1 to Day 92 (before switch-over).

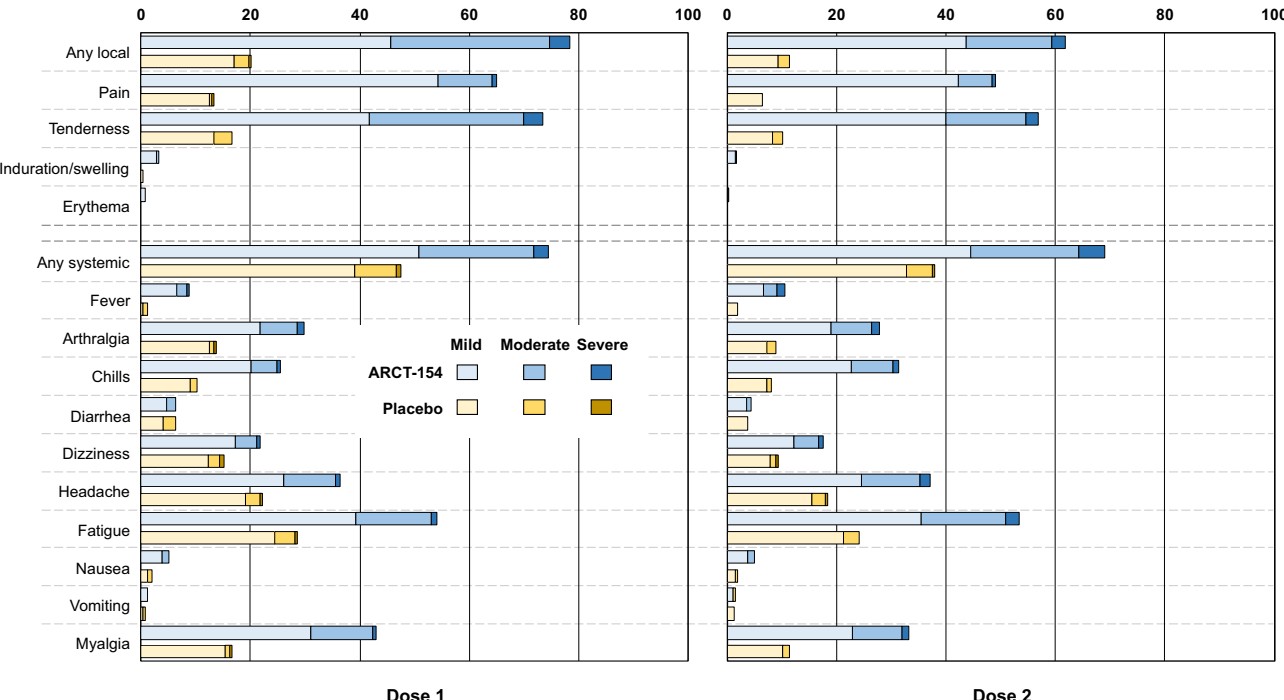

**Fig. 3 | Solicited reactogenicity during 7 days after Doses 1 and 2 of ARCT-154 or placebo** in the combined phase 1, 2 and 3a studies, with highest severity indicated as mild (Grade 1), moderate (Grade 2) or severe (Grade 3).

**Percentage of each group reporting adverse events by severity**

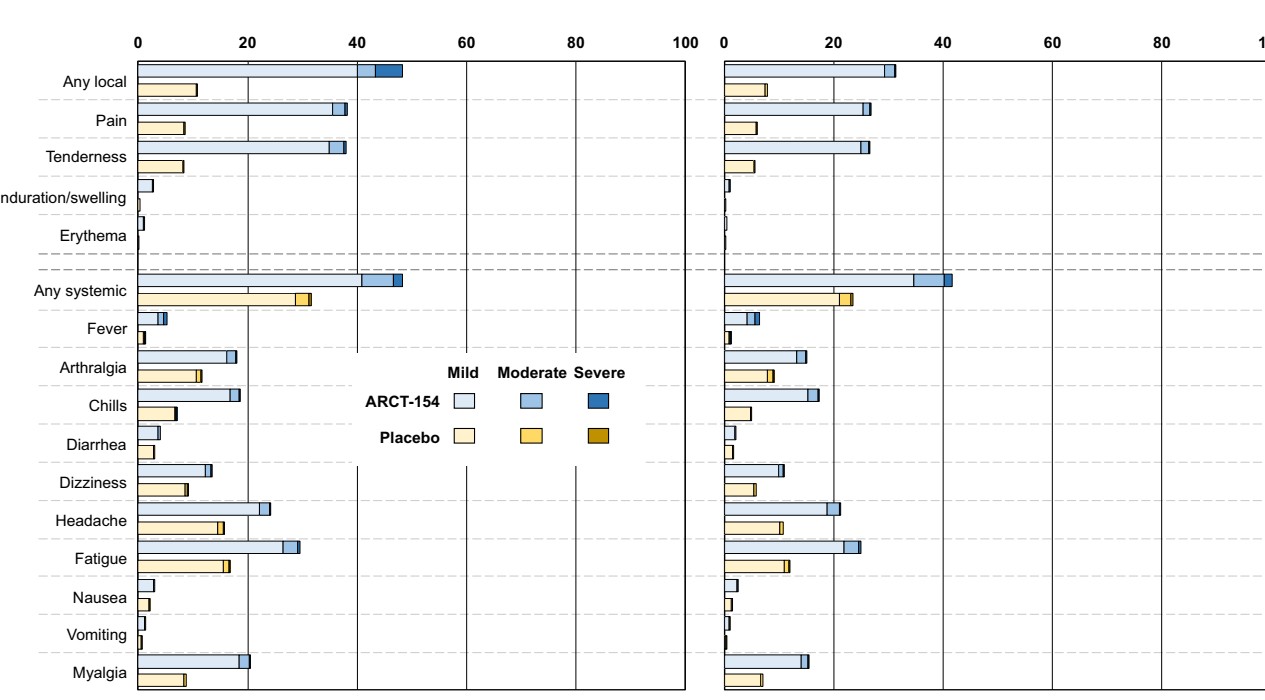

**Fig. 4 |** Solicited reactogenicity during 7 days after Doses 1 and 2 of ARCT-154 or placebo in the phase 3b study, with highest severity indicated as mild (Grade 1), moderate (Grade 2) or severe (Grade 3).

prespecified success threshold of 30%. Secondary analyses showed high efficacies against severe COVID-19 (95.3% [80.5–98.9]) and death due to COVID-19 (86.5% [-7.4–98.3]). Efficacy against any severity of COVID-19 was similar in male and female participants (Table 3). Efficacy against severe COVID-19 was 100% in healthy 18–59-year-olds and 91.9% (37.9–98.9) in "at-risk" participants in the same age group. In adults aged 60 years or older efficacy was 54.3% (28.2–70.9) against COVID-19 of any severity and 94.4% (58.2–99.3) against severe COVID-19.

In the mITT set 537 cases were analyzed to identify the responsible SARS-CoV-2 variant, revealing 477 (88.8%) were Delta (B.1.617.2) variant (164 and 313 in vaccine and placebo groups), two were Alpha (both placebo) one was Beta (placebo) and two were Omicron (one each vaccine and placebo) variants; viral variant was not determined in 55 (10.2%) cases (Supplementary Table 5). When assessed in only mITT cases in which Delta (B.1.617.2) was the identified variant two doses of ARCT-154 had 49.8% (95% CI: 39.3–58.4) efficacy against COVID-19 of any severity and 94.3% (57.4–99.2) against severe COVID-19 (Supplementary Table 6).

An additional 52 COVID-19 cases observed from Day 1 through Day 35, including three severe cases, were adjudicated to be eligible for secondary analyses of efficacy in the phase 3b Intention to Treat (ITT) set in which efficacy after any dose of ARCT-154 from Day 1 to Day 92 was similar to the mITT analyses (Table 3, Supplementary Fig. 2); observed efficacy against COVID-19 of any severity was 56.6% (95% CI: 49.0–63.1) and 95.6% (81.5–98.9) against severe COVID-19. Finally, calculated efficacies against any severity of COVID-19 in the pooled phase 1, 2, and 3a participants were consistent with the observations in the phase 3b study as efficacy against any severity of COVID-19 from Day 36 to Day 92 (mITT) was 56.3% (95% CI: 18.2–76.7) and from Day 1 to Day 92 (ITT) was 58.9% (95% CI: 23.8–77.8) in that population (Supplementary table 7).

## Discussion
As new variants of SARS-CoV-2 virus continue to emerge approved mRNA vaccines have been found to have lower effectiveness estimates

compared with the efficacy rates measured in their pivotal studies[6–8]. We achieved the primary objectives of phases 1, 2, and 3a of this integrated study, successful demonstrating acceptable safety and reactogenicity, and immunogenicity of ARCT-154, with 95·9% seroconversion for neutralizing antibodies against the SARS-CoV-2 D614G variant. Demonstration of safety in these initial phases allowed recruitment into the larger phase 3b study population in which we show that two doses of ARCT-154 had a vaccine efficacy (VE) of 56.6% (95% CI: 48.7–63.3) against any severity of COVID-19 and more notably 95.3% (80.5–98.9) against severe COVID-19 in a background of predominantly Delta (B.1.617.2) SARS-CoV-2 variant which caused 88.8% of infections where the variant was identified. This is the first published demonstration of the clinical efficacy of an sa-mRNA vaccine. ARCT-154 had > 90% efficacy against severe COVID-19 in those at risk for severe disease, those aged over 60 years, and adults from 18 to 59 years of age with underlying co-morbidities. Efficacy against death due to COVID-19 was 86.5% (−7.4–98.3); the wide confidence limits reflecting that there were only 10 deaths, one vaccinee and nine placebo recipients.

Efficacy of the first approved mRNA vaccines was demonstrated at a time when relatively low proportions of the study populations had been exposed to the circulating SARS-CoV-2 virus, and efficacy was measured against the prototype Wuhan-Hu-1 strain or one of the first variants to emerge which had only minor changes in the antigenic structure of the S protein target of these vaccines, rather than Delta (B.1.617.2). Hence, the original efficacy estimates of these vaccines against COVID-19 illness, including severe disease, were higher than that observed for ARCT-154, e.g., 95% (95% CI: 90.3–97.6) for BNT162b2[20] and 94·1% (89.3–96.8) for mRNA-1273[21]. However, the effectiveness of these first approved mRNA vaccines was observed to decline against emerging variants of concern, exacerbated by waning immunity following the initial vaccination series[5–8]. The effectiveness of a completed primary vaccination series of authorized COVID-19 vaccines against infection by the Delta (B.1.617.2) variant has been

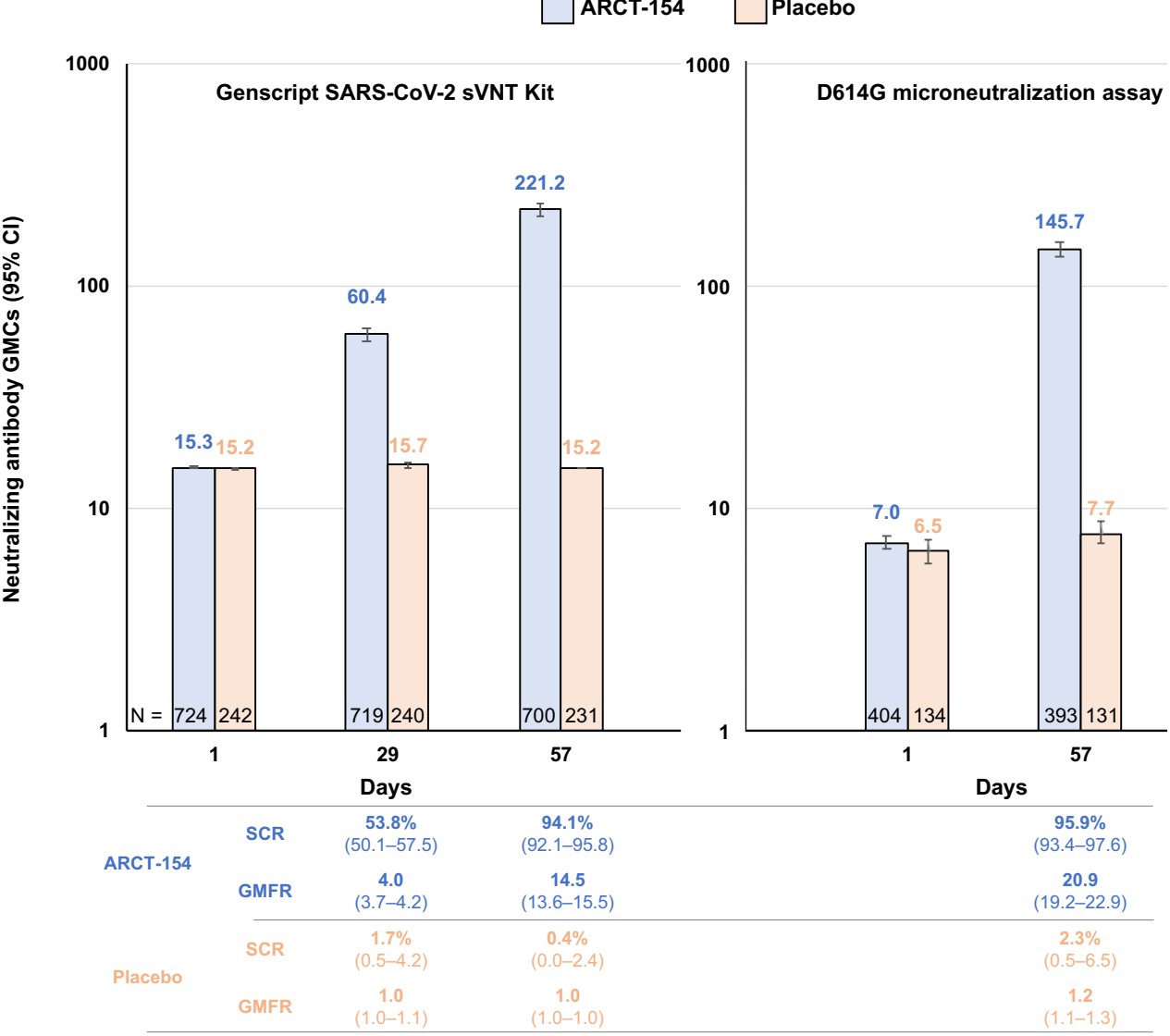

**Fig. 5 | Neutralizing antibody responses before and after the two study vaccinations measured using two different assays.** Responses shown Geometric mean concentrations of neutralizing antibodies (with 95% CI bars) and with seroconversion rates (SCR) and geometric mean-fold rises (GMFR) from baseline indicated below. Values shown are for N participants (indicated in each column) at each timepoint.

shown to range between 46% and 91%, and from 47% to 97% against severe COVID-19 disease due to Delta (B.1.617.2) variant[22].

The lower VE of the ARCT-154 vaccine we observed may be a consequence of the definition of COVID-19 disease used in the trial, being based on presence of a single symptom in combination with positive RT-PCR. This allows the inclusion of a significant number of mild and marginally symptomatic cases in the analysis, and it is acknowledged that the effectiveness of COVID-19 vaccination increases with the severity of COVID-19 and decreases with time since vaccination. Effectiveness of a 3-dose series of BNT162b2 vaccine against mild, moderate and severe COVID-19, caused by the Omicron variant within 31 to 60 days after the last vaccination, was 7.9% (95% CI: 2.3–13.1), 49.2% (95% CI: 46.8–51.4) and 76.4% (95% CI: 72.4–79.8), respectively[23]. It was notable that ARCT-154 was much more efficacious against severe COVID-19 than disease of any severity. As such, the case definition used for primary efficacy analysis has a significant impact on the efficacy point estimate. Clinical studies with other vaccines have generally used a 'more symptomatic' definition of COVID-19 disease (a positive RT-PCR in combination with at least two systemic symptoms or at least one the respiratory signs or symptom).

Safety and reactogenicity in all phases indicate the vaccine is well tolerated, with mainly mild or moderate adverse events, most of which were transient local reactions. The most frequent solicited systemic adverse events were transient fatigue, myalgia, headache, and chills which resolved more quickly than the local reactions. All adverse events were more frequent after vaccine than placebo. Although trials were not identical in design so not directly comparable, reporting methods are sufficiently similar to allow us to note that overall systemic AEs and local reactions were similar or less frequent in recipients of ARCT-154 than licensed mRNA vaccines[24]. Although differences in reactogenicity might also be associated with cultural differences in the reporting of subjective adverse events when used a booster dose in mRNA-primed adults ARCT-154 had an almost identical reactogenicity profile as the mRNA vaccine, BNT162b2, when observed in a head-to-head comparison in Japan adults[25].

Assessment of vaccinee immunogenicity against SARS-CoV-2 ancestral strain using the sVNT assay showed 94.1% seroconversion in vaccinees four weeks after the second dose, which was confirmed using the validated microneutralization assay which showed 95.9% seroconversion. There is currently no serologic correlate for

**Table 3 | Primary and key secondary vaccine efficacy (VE) endpoints in the modified ITT (mITT) analysis population[a]**

| | Total no. of persons | Follow-up in person-yrs | No. with event | Total no. of persons | Follow-up in person-yrs | No. with event | Vaccine efficacy (95% confidence intervals) |
|---|---|---|---|---|---|---|---|
| **Primary VE endpoints in the mITT population[a,b]** | **ARCT-154** | | | Placebo | | | |
| **Any severity** protocol-confirmed COVID-19 (Days 36–92) | 7787 | 1131.7 | 200 | 7723 | 1100.6 | 440 | 56.6% (48.7–63.3) |
| **Key secondary VE endpoints in the mITT population[b]** | | | | | | | |
| **Severe** protocol-confirmed COVID-19 (Days 36–92) | 7787 | 1148.2 | 2 | 7723 | 1134.8 | 41 | 95.3% (80.5–98.9) |
| **Death** due to protocol-confirmed COVID-19 (Days 36–92) | 7787 | 1148.4 | 1 | 7723 | 1138.4 | 9 | 86.5% (−7.4–98.3) |
| **Any severity** protocol-confirmed COVID-19 (Days 36–92) according to gender, age or at-risk status[c] | | | | | | | |
| **Females** | 3970 | 571.2 | 115 | 3930 | 557.0 | 234 | 53.3% (41.6–62.6) |
| **Males** | 3817 | 560.4 | 85 | 3793 | 543.6 | 206 | 61.1% (49.9–69.8) |
| **Healthy ≥ 18 to < 60-year-olds** | 3704 | 535.4 | 119 | 3701 | 5265.4 | 235 | 50.8% (38.7–60.6) |
| **"At risk" ≥ 18 to < 60-year-olds[c]** | 2719 | 396.4 | 53 | 2690 | 383.6 | 148 | 66.5% (54.2–75.5) |
| **≥ 60-year-olds** | 1364 | 199.8 | 28 | 1322 | 190.5 | 57 | 54.3% (28.2–70.9) |
| **Severe** protocol-confirmed COVID-19 (Days 36–92) according to age or at-risk status[c] | | | | | | | |
| **Healthy ≥ 18 to < 60-year-olds** | 3704 | 545.0 | 0 | 3701 | 545.4 | 12 | 100% (32.5–0.0) [d] |
| **"At risk" ≥ 18 to < 60-year-olds[3]** | 2719 | 401.2 | 1 | 2690 | 395.6 | 12 | 91.9% (37.9–98.9) |
| **≥ 60-year-olds** | 1364 | 202.1 | 1 | 1322 | 193.9 | 17 | 94.4% (58.2–99.3) |
| **Key secondary VE endpoints in participants with no evidence of infection at baseline (ITT)[e]** | | | | | | | |
| **Any severity** protocol-confirmed COVID-19 (Days 1–92) | 8056 | 1942.3 | 215 | 8043 | 1911.5 | 477 | 56.6% (49.0–63.1) |
| **Severe** protocol-confirmed COVID-19 (Days 1–92) | 8056 | 1961.9 | 2 | 8043 | 1952.3 | 44 | 95.5% (81.5–98.9) |
| **Death** due to protocol-confirmed COVID-19 (Days 1–92) | 8056 | 1962.1 | 1 | 8043 | 1956.6 | 9 | 86.9% (−3.7–98.4) |

[a]VE was calculated by 1-HR from Cox regression adjusting for risk group and region of study site.
[b]mITT population included all those who received both study doses (ARCT-154 or placebo) and who had no evidence of SARS-CoV-2 infection from Day 1 to Day 36.
[c]At-risk based on presence of predefined medical conditions known to be associated with risk of severe COVID-19 (see ref. 32)
[d]if the number of events was equal to 0 in one of the groups, VE was calculated according to[33] and[34]. Note, upper 95% confidence limit was not estimable.
[e]No evidence of infection on basis of negative serological test for SARS-CoV-2 nucleocapsid protein - this analysis was performed on the ITT population.

protection by anti-SARS-CoV-2 antibodies, but neutralizing antibody levels are highly predictive of immune protection[26]. This study probably represents the last opportunity to assess the efficacy of ARCT-154 in a SARS-CoV-2 naïve population as the global pandemic has resulted in most people having some exposure either through vaccination, natural infection, or both leading to hybrid immunity[27]. Due to the high level of vaccination coverage with waning immunity, the current need for COVID-19 vaccines is to re-establish population immunity to protect against the emerging variants still causing outbreaks[4]. As such, ARCT-154 is most likely to be used as a booster dose, rather than for primary immunization, to enhance and broaden the level of immunity against circulating variants. A parallel study in Japan has shown that in adults fully immunized with mRNA vaccines, mainly BNT162b2, as the primary vaccine, the immune response to a booster dose of ARCT-154 was superior to that of a booster dose of BNT162b2 when measured as neutralizing antibodies against Wuhan-Hu-1 and the Omicron 4/5 subvariant[25]. Further, the persistence of the response to ARCT-154 was better than to BNT162b2 up to 6 months after boosting[28]. The licensed vaccines have now been adapted to reflect the changing epidemiology of SARS-CoV-2 including formulations based on the S-protein of the latest Omicron XBB.1.5 variant which are the currently recommended vaccines in the United States[29].

Preclinical animal studies with the same proprietary technology as ARCT-154 have shown that it provides a durable immune response that includes the induction of neutralizing antibodies, and the activation of cell-mediated responses including CD4 + T cell interferon-γ and interleukin-4 secretion, antigen-specific CD8 + T cell responses, and an anti-spike protein IgG2: IgG1 ratio indicating a Th1-type dominant response[30]. It remains to be seen whether ARCT-154 induces similar responses in humans. In adults from 18-75 years of age Szubert et al. found 1 μg and 10 μg doses of an investigational lipid encapsulated SARS-CoV-2 sa-RNA vaccine induced neutralizing and anti-spike-IgG antibodies although there was only a modest correlation between the two measures[31–34].

The study has recently completed and some of the limitations in this report, such as 12-month safety results and further investigations into the immunogenicity are being analyzed and will be disclosed in separate papers. Notably, long-term safety up to one year after vaccination and further investigations of the humoral and cellular immunogenicity to elucidate the nature of the immune response. Follow-up will also allow assessment of the durability of this response and levels of cross-neutralizing antibodies against newly emerged variants. Limited cross-neutralization data are available, but results from the already mentioned booster study in Japan show that ARCT-154 induces a superior cross-neutralizing response against Omicron BA.4/5 than a comparator mRNA vaccine both in magntiude[25] and persistence[28]. Updated formulations will also be required based on the latest variants, such as Omicron XBB.1.5 as already mentioned. If sa-mRNA vaccines do provide equivalent protective efficacy as the licensed mRNA vaccines, but with a lower amount of mRNA, there is potential to decrease the manufacturing cost per dose or to allow production of more doses which may be important factors in a future pandemic[30].

This first demonstration of the clinical efficacy of the ARCT-154 sa-mRNA vaccine against COVID-19, together with acceptable safety and

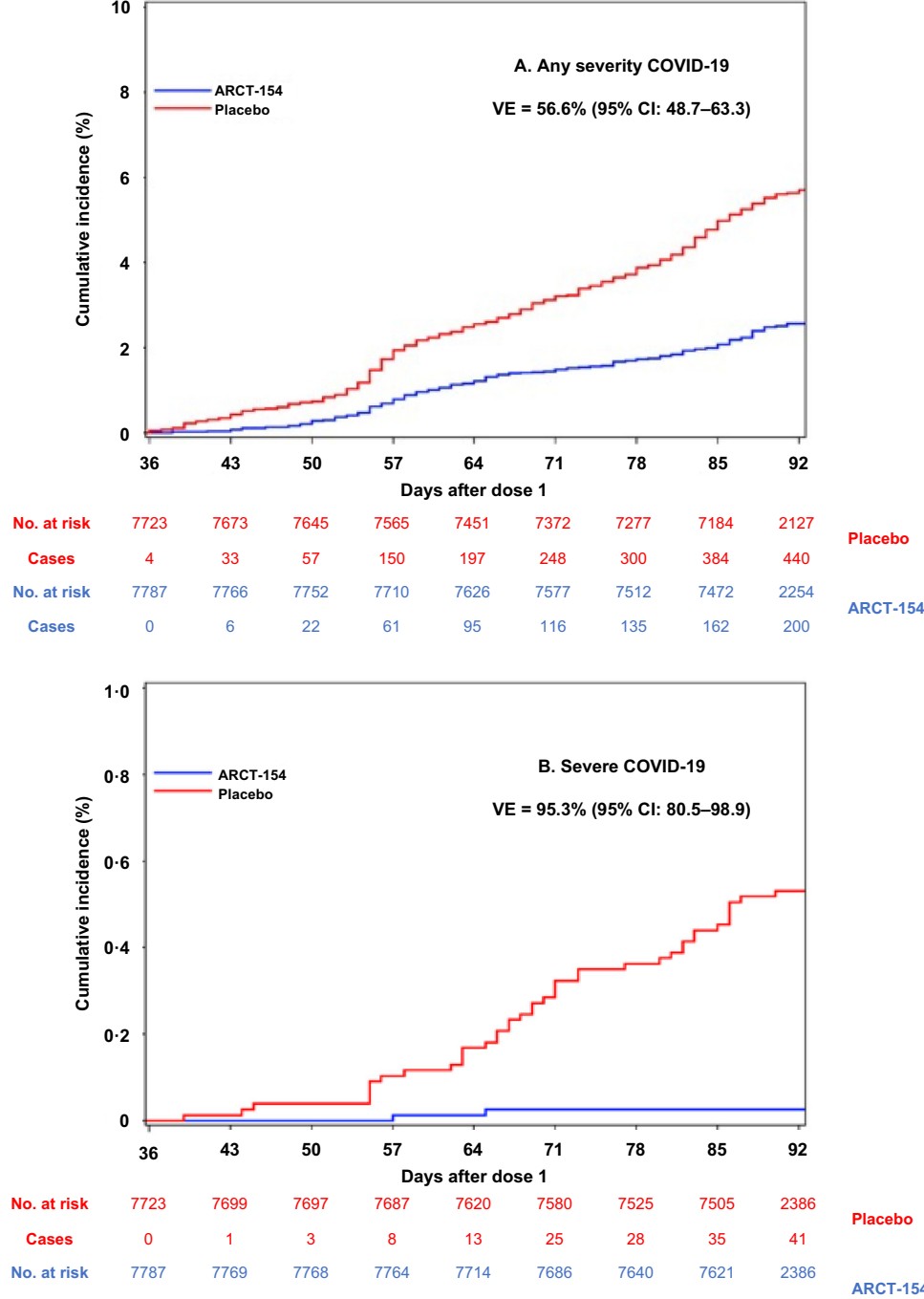

**Fig. 6** | Cumulative incidence curves of COVID-19 of any severity (**A**), and severe COVID-19 (**B**) in vaccine and placebo groups from Day 36 (per protocol).

reactogenicity in a large study population, establishes the potential of sa-mRNA vaccines for future clinical use, and complements the other study that showed boosting with ARCT-154 provides superior immunogenicity against Omicron than an mRNA vaccine[25,28]. This supports the further development of sa-mRNA vaccines to augment the armamentarium against future COVID-19 outbreaks.

## Methods

The protocol was approved by the ethics committee (EC) of each of the 16 study centers and the Vietnam National EC in Biomedical Research and Ministry of Health. All participants provided informed consent before enrolment, and the study was conducted in accordance with the ethical principles of Good Clinical Practice, the Declaration of

Helsinki and International Council for Harmonisation (ICH E6R2), and applicable local regulatory and bioethics requirements. Primary objectives of phase 1, 2, and 3a studies were to assess in comparison with placebo the safety and reactogenicity of two ARCT-154 doses administered four weeks apart, and the immunogenicity four weeks after the second dose. Primary objectives of the phase 3b study were to assess the safety and reactogenicity of two doses of ARCT-154 and their efficacy against COVID-19 disease from 7 (Day 36) to 63 (Day 92) days after the second vaccination. An exploratory assessment of efficacy from Day 1 to Day 92 was done in all participants who had received at least one dose of ARCT-154 in any phase of the study. An independent data and safety monitoring board (DSMB) had full oversight over the study, including assessment of blinded data on safety and confirmed

COVID-19 cases, and made recommendations to continue enrolment or the study.

## Participants and study design

Study designs for individual study phases are illustrated in Fig. 1. In phase 1 eligible participants were healthy adults > 18 to <60 years of age; in phases 2/3a/3b eligible participants were adults > 18 years of age. Enrolled volunteers in phases 1/2/3a were randomized 3:1 and those in phase 3b 1:1 to receive ARCT-154 or placebo. Phase 3b included volunteers at increased risk of severe COVID-19 due to their comorbidity status[31] or being ≥ 60 years old, and randomization included stratification of these at-risk participants. Phase 1 participants were recruited and treated first, parallel enrolment for phases 2 and 3a was only allowed after the DSMB and Vietnam MoH had reviewed all safety data collected up to 7 days after the second vaccination (Day 36) in phase 1. Similarly, enrolment for phase 3b was only approved following review of all safety data collected through Day 7 following the first dose in phases 2 and 3a.

Other than the above age restrictions, eligible participants were male or female adults who could consent to participate, agreed to comply with all required study visits and procedures, and were willing to provide required blood and nasal swab samples. Major exclusion criteria were evidence of an acute infection at the time of enrolment, pregnancy or breastfeeding, previous COVID-19 infection (including a positive result of RT-PCR), close contact with a person known to be infected with SARS-CoV-2 or any known history of anaphylactic reactions to vaccines. Detailed exclusion criteria are shown in Supplementary Table 2.

At enrolment volunteers were allocated to ARCT-154 or placebo groups using an interactive response technology (IRT) system which provided a unique identifying study code and the allocated study intervention for each participant. Codes were accessible only to unblinded study personnel who prepared and administered the vaccine/placebo but played no other role in the study. All other study personnel and participants were blinded to study allocation. A nasal swab was collected for SARS-CoV-2 testing by RT-PCR at screening on or within 5 days of Day 1. On Day 1, after a baseline blood draw for testing for SARS-CoV-2 nucleocapsid-specific antibodies and negative urine or blood pregnancy test for females of child-bearing potential, the first assigned vaccine or placebo injection was administered; a second dose was given in the same manner on Day 29. To ensure all participants received immunization against COVID-19 there was as switchover at Day 92 when placebo recipients from all phases were offered ARCT-154 as two doses four weeks apart. Vaccinees from the different phases received either a third dose of ARCT-154 or two doses of placebo. This report only presents data acquired up to Day 92, data from the switchover will be presented separately.

## Vaccine

ARCT-154 consists of a replicon based upon Venezuela equine encephalitis virus in which RNA coding for the virus structural proteins has been replaced with RNA coding for the full-length spike (S) glycoprotein of the SARS-CoV-2 D614G variant, encapsulated in lipid nanoparticles. 100 µg active ingredient, stored in vials at -20 °C or lower, was dissolved in 10 mL sterile saline immediately before use and 0.5 mL doses containing 5 µg were administered by intramuscular injection in the deltoid. Placebo was sterile saline.

## Safety and reactogenicity

After 30 minutes monitoring for any immediate reactions, all participants completed electronic or paper study diaries for 7 days starting on the day of each study injection. Diaries solicited local reactions (injection site erythema, pain, induration/swelling, and tenderness) and systemic adverse events (AEs; arthralgia, chills, diarrhea, dizziness, fatigue, headache, myalgia, nausea/vomiting and fever). Unsolicited AEs were recorded up to 28 days after each vaccination. Any adverse event leading to discontinuation or withdrawal from the study, any medically attended adverse event (MAAE) or serious adverse event (SAE) was to be documented for one year of follow-up after the completion of the initial vaccination series. Here we present general safety data including MAAEs, SAES and withdrawals up to six months (Day 210). Participants were contacted through weekly telephone calls to ensure compliance with completing the study diaries, which were collected on Days 8 and 36, 7 days after each vaccination, and at a follow-visit on Day 57. Adverse event data was entered into the case report form, and the causal relationship of events was established by the reporting investigator.

## Immunogenicity

Sera for immunogenicity analyses were collected on Days 1, 29 and 57. The primary immunogenicity objective was the response at Day 57 in all sera available from eligible phase 1/2/3a participants as measured at the Vietnamese National Institute of Hygiene and Epidemiology Laboratory using the SARS-CoV-2 Surrogate Virus Neutralization Test (sVNT) kit (GenScript, Piscataway, NJ, USA). This kit is a functional enzyme-linked immunosorbent assay for qualitative or semi-quantitative detection of antibodies (Nabs) that block the binding of SARS-CoV-2 to the human ACE2 receptor of host cells. Antibodies were expressed as group geometric mean concentrations (GMC), seroconversion rates (SCR), and geometric mean fold rises (GMFR) from Day 1. Results were expressed in units per mL (U/mL) calibrated with the WHO standard serum.

To confirm observations from the exploratory assay, immunogenicity was also assessed in a validated 293T-ACE2 cell-based microneutralization assay by the Pharmaceutical Product Development Bioanalytical Laboratory (PPD, Richmond, VA, USA). This measured neutralizing antibodies against SARS-CoV-2 in sera from Days 1 and 57 from all phase 1 participants, the first 150 samples from phase 2, and randomly selected samples from the other participants in phase 2, and all available samples from phase 3a.

## Evaluation of participants with suspected COVID-19

For efficacy assessments, participants with suspected COVID-19 were evaluated for the presence of potential symptoms and clinical signs of COVID-19 including fever, chills, cough, shortness of breath or difficulty breathing, fatigue, muscle or body aches, headache, new loss of taste or smell, sore throat, congestion or runny nose, nausea, vomiting or diarrhea. Any of these symptoms occurring after 3 days post-vaccination triggered COVID-19 diagnostic testing. Where possible, participants visited their respective study clinic where nasal swabs were taken for RT-PCR, with documentation of medical history and medications taken. A protocol-defined COVID-19 case had to have virological confirmation (by RT-PCR) of SARS-CoV-2 and at least one of the symptoms or clinical findings listed above.

Case definitions for evaluations of COVID-19 and severe COVID-19 were based on US FDA recommendations in line with similar clinical trials, which for severe COVID-19 included any of the following: acute pulmonary, cardiac, renal, hepatic, or neurologic dysfunction; shock; death; or admission to an intensive care unit (Supplementary table 3). All suspected COVID-19 cases underwent blinded tiered review by an independent Event Adjudication Committee (EAC) composed of clinical experts experienced in the diagnosis, care, and treatment of COVID-19. The EAC reviewed blinded data from each case and concluded on whether the case met the protocol-defined COVID-19 case criteria, and severity according to the US FDA and WHO classifications. Only virologically confirmed, protocol-defined cases adjudicated by the EAC are included in the primary vaccine efficacy (VE).

## Statistical analysis

Primary safety endpoints were evaluated in the Safety Analysis Set (SAS; all participants who received any study injection) and Reactogenicity Analysis Set (RAS; all participants who received any study injection and provide at least one diary report). Statistical analysis of safety and reactogenicity data was descriptive with frequency and percentage for participants analyzed according to study group.

Primary immunogenicity analysis in the Immunogenicity Analysis Set (IAS) included all participants who received both assigned study injections by the evaluated timepoint with no evidence of prior SARS-CoV-2 infection at Day 1 (i.e., were seronegative for N-antibody) and at least one valid post-vaccination immunogenicity assay result. GMCs were calculated as the mean of log-transformed results and then exponentiating the mean (in order to present the results on the original scale). GMFR was calculated as the mean of the difference after log-transformed results (post baseline minus baseline) and exponentiating the mean. Two-sided 95% CI for GMCs and GMFRs were obtained by taking log-transformation of the antibody results; the 95% CI was calculated based on Student's t-distribution for the mean difference, then exponentiating the confidence limits. Seroconversion was defined as 4-fold increase in titer from baseline and its two-sided 95% CI was calculated using the Clopper-Pearson method.

The primary efficacy objective was assessed in the modified Intention to Treat (mITT) set composed of all participants who received both assigned study injections and had no evidence of SARS-CoV-2 infection on Day 1 and up to Day 36, 7 days after the second study injection. The first primary endpoint was defined as the first occurrence of confirmed, protocol-defined COVID-19 with onset between Days 36 and 92 inclusive. For the overall primary efficacy objective of the study, the null hypothesis was that the vaccine efficacy (VE) of ARCT-154 to prevent COVID-19 was ≤ 30% (i.e., $H0^{efficacy}$: VE ≤ 0.3). Vaccine efficacy was calculated from 1-hazard ratio, where the hazard ratio (HR) and 95% CI are estimated by Cox proportional hazard regression. The primary efficacy objective would be met if the lower limit of the 95% CI for VE exceeded 30%; a total of 372 COVID-19 cases were needed to provide approximately 90% power to detect a 50% reduction in hazard rate (50% VE). Factors used as covariates in Cox proportional hazard regression included: Risk group: ≥ 18 to <60 years and "healthy", ≥ 18 and <60 years and "at risk" and ≥ 60 years and study site region. If the primary efficacy objective was met, following a hierarchical approach, the null hypothesis that the vaccine efficacy to prevent occurrence of confirmed severe COVID-19 was ≤ 0% (i.e., $H0^{efficacy}$: $VE_{severe}$ ≤ 0) was also tested. A secondary efficacy assessment was done in the ITT set, comprising all participants who received at least one study injection, in which the secondary endpoint was the occurrence of confirmed, protocol-defined COVID-19 with onset at any time after the Dose 1 up to Day 92, inclusive.

## Reporting summary

Further information on research design is available in the Nature Portfolio Reporting Summary linked to this article.

# Data availability

After the final study report is prepared, including the 12-month safety follow-up period, the data generated in this study will be made available to suitably qualified scientific researchers who make a request to the senior investigator or study sponsor with a appropriate protocol for a valid research project.

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

## Acknowledgements

The authors are grateful to the study staff at Hanoi Medical University, Pasteur Institute Hochiminh City, and Vietnam Military Medical Uni-versity for assisting with study conduct; Hang Thi Vi, Trang Thu Hoang, Huong Vu Quynh Ngo, Ha Thai Pham, and Ngan Thi Le (Vietnam Biocare Biotechnology Jointstock Company) for contributing to study oversight; Lan Quynh Phan (Vinmec Healthcare System) for investigational product oversight; Anh-Tien Ngo (Hi-Tech Center, Vinmec Healthcare System) for sample processing and storage; Prof. Duc-Anh Dang, Thi-Khanh-Hang Le, NIHE and Vu-Tien Phan (Pasteur Institute Nha Trang) for immunogenicity assays; VietStar Biomedical Research for participating in study conduct and management. Hongfan Jin co-ordinated laboratory testing, Cindy Fisher, Ye Zhang and Pamela Resch participated in dis-cussions with regulatory authorities regarding the protocol and devel-opment plan, and Deep Patel, Mukunda Krishna, Charles Cabral, and Claudia Averbuj provided expert support in drug product development, manufacturing and supply. We thank Keith Veitch (keithveitch commu-nications, Amsterdam, The Netherlands) for editorial assistance in the preparation of the manuscript. The study was co-funded by Vinbiocare Biotechnology Joint Stock Company (Hanoi, Vietnam) and Arcturus Therapeutics Inc. (CA, USA), with no particular grants accorded to any specific researchers.

## Author contributions

N.T.H., X.-H.N., L.T.L.T., V.T.N., S.G.H. and K.L. participated in the design, protocol development and conduct of the study. N.T.H., S.G.H., B.G. and I.S. participated in verifying the underlying data reported in the manu-script. J.M.E. reviewed the study results and provided critical review of the manuscript. S.P., B.S., S.S., Q.R., B.C., P.C. and B.L. participated in vaccine development, optimization, manufacturing control, and vaccine release for clinical use. B.G., I.S., N.T.H. and P.C. oversaw and partici-pated in the data analysis plan, data analysis and manuscript prepara-tion. V.T.N., X.-H.N., K.M. and D.B. had overall management of the study. X.-H.N., N.T.H.N., H.-S.V. and R.S. oversaw laboratory testing and ana-lyses. L.T.L.T., A.T.V.L. and A.N.N. oversaw study operations. V.T.T., A.T.V.P., T.Vu.N., L.P.T., H.N.P., M.V.C., M.T.N.D., Q.V.T., Q.C.L., T.T.N., V.T.T.L., Q.D., L.V.N. and T.Van.N. oversaw study conduct at sites.

## Competing interests

S.H., S.P., B.S., S.S., Q.R., B.C., B.L., K.L., D.B., K.M., R.S., B.G. and I.S. were all full-time employees of the vaccine manufacturer and study sponsor, Arcturus Therapeutics, Inc., at the time of the study. T.T.L.L. and N.T.V. are employees of the vaccine licensee, Vietnam Biocare Bio-technology Joint Stock Company. Other authors declare no competing interests.

## Additional information

Nhân Thị Hồ[1], Steven G. Hughes[2], Van Thanh Ta[3], Lân Trọng Phan[4], Quyết Đỗ[5], Thượng Vũ Nguyễn[4], Anh Thị Văn Phạm[3], Mai Thị Ngọc Đặng[3], Lượng Viết Nguyễn[5], Quang Vinh Trịnh [ID][3], Hùng Ngọc Phạm[5], Mến Văn Chử[5], Toàn Trọng Nguyễn[4], Quang Chấn Lương[4], Vy Thị Tường Lê[4], Thắng Văn Nguyễn[5], Lý-Thi-Lê Trần[6,7], Anh Thi Van Luu[7], Anh Ngoc Nguyen[7], Nhung-Thi-Hong Nguyen [ID][1], Hai-Son Vu[1], Jonathan M. Edelman[8], Suezanne Parker[2], Brian Sullivan [ID][2], Sean Sullivan[2], Qian Ruan[2], Brenda Clemente[2], Brian Luk[2], Kelly Lindert[2], Dina Berdieva[2], Kat Murphy[2], Rose Sekulovich[2], Benjamin Greener[2], Igor Smolenov [ID][2], Pad Chivukula [ID][2], Vân Thu Nguyễn[7] & Xuan-Hung Nguyen [ID][1,6,9] ✉

[1]Vinmec-VinUni Institute of Immunology, Vinmec Healthcare System, Hanoi, Vietnam. [2]Arcturus Therapeutics, Inc, San Diego, CA, USA. [3]Hanoi Medical University, Hanoi, Vietnam. [4]Pasteur Institute, Ho Chi Minh City, Vietnam. [5]Vietnam Military Medical University, Hanoi, Vietnam. [6]Hi-tech Center, Vinmec Healthcare System, Hanoi, Vietnam. [7]Vietnam Biocare Biotechnology Jointstock Company, Hanoi, Vietnam. [8]CSL Sequiris Inc, New Jersey, USA. [9]College of Health Sciences, Vin University, Hanoi, Vietnam. ✉e-mail: v.hungnx1@vinmec.com

