## [Peer Review File · Nature Communications]

Safety and immunogenicity and efficacy of the self-amplifying mRNA ARCT-154 COVID-19 vaccine; pooled phase 1, 2, 3a and 3b randomized, controlled trialsReviewers' Comments:

Reviewer #1:

Remarks to the Author:

The authors conducted a comprehensive set of clinical trials evaluating the safety, immunogenicity and efficacy of ARCT-154, a VEEV replicon based self-amplifying RNA (saRNA) vaccine encoding the full-length spike protein of SARS-CoV-2 with D614G, as the primary series in a population mostly without prior SARS-CoV-2 infection. This is the saRNA-based COVID vaccine with the best reacto, immunogenicity and efficacy profiles and kudos to the authors for conducting a nice set of trials for the great info about this vaccine candidate. Given the extensive clinical and observational studies for BNT162b2 and mRNA-1273, however, accuracy needs to be maintained with comparison is described in the manuscript:

1. Line 230-234: Given the trial is measuring VE against Delta, it's not fair to reference a paper describing VE of the two commercial COVID vaccines against Omicron in order to show case the ARCT-154. There is no VE against Omicron/BA.1 or any durability data from ARCT-154 in this manuscript, so it's critical to stay in course when comparing this vaccine with the others.

2. In numerous observational studies, VE (effectiveness) of the primary series of the two commercial COVID vaccine against Delta infection or symptomatic infection has been shown to be above 70%. It's worth to reference and discuss those studies in the Discussion. This will not only help the readers to understand the differences of COVID shots but also provide valuable insights to the sponsor of this study on prospect of this candidate product.

Reviewer #2:

Remarks to the Author:

- The authors present safety and per protocol efficacy data from Phases 1, 2, 3a and 3b of an ongoing study evaluating a candidate saRNA vaccine which provides up to three months follow-up, so too short a duration to assess whether these results are noteworthy in the context of licensed COVID19 vaccines
- Nonetheless, an impressive preliminary clinical trial dataset with results that are worthy of publication subject to clarification.
- I am not a virologist, but the assays used to assess seroconversion, defined as 4-fold change in GMT from baseline, are both validated and were conducted in independent laboratories which I imagine are subject to inspection of their quality systems.
- Assessment of clinical endpoints follows FDA guidance and was subject to adjudication by independent clinical experts who were blind to allocation.
- Consort and Tables 2 and 4 – I found it difficult to follow the numbers. What exactly to the authors mean by 'exposed' in the consort? Numbers do not match with second dose in Table 1. 2,295 (14%) participants withdrew from the phase 3b study and 79 (8%) from phases 1/2/3a; this is given as the main reason for 'study treatment discontinuation' but makes no sense to me when viewed alongside the numbers in table 1 for compliance with second dose. So what do they mean by 'study treatment discontinuation'? Were they lost after the second dose, but before the 92 day cut-off? If so, readers need to know as this amounts to 14% which seems high to me as a loss from the main analyses and a point to be mentioned in the discussion with respect to limitations. It seems to me they have included everyone who entered the mITT, but what I would like to know is how many participants do they know the clinical status of at day 92? It is this number and proportion that gives me confidence in the results – more so than person years and rates which can mask loss of those more vulnerable to severe disease and therefore endpoints.
- The safety profile presented is similar to licensed mRNA vaccine with more reactogenicity than recombinant protein vaccines. The candidate vaccine was sufficiently well tolerated for almost all participants to return and receive the second vaccine, and reactogenicity appeared to diminish as the size of the trial and age of participant increased.
- Participants had to be naïve to vaccines, and report no history of COVID-19 or a positive test. Nonetheless they could have had asymptomatic natural infection, or a symptomatic infection that they

chose not to report. From Table 4 it seems anti-N was assessed for all participants to generate the dataset with no evidence of infection at baseline. and remarkably few had these antibodies if the heading for so include the n(%) with anti-nucleocapsid antibodies and whether this made a difference to inclusion in the analysis sets eg were they excluded from the mITT dataset for 3b?

- The immunogenicity results in table 3 support the clinical efficacy in table 4 with high rates of seroconversion after 2 doses and a strong effect for protection against severe COVID19 respectively. This interpretation is subject to my caveats above about the numbers lost to follow-up.
- Good to see the clinical effectiveness is similarly high in older age-groups and those with risk factors.
- The authors note that the strength of this candidate is in protection against severe disease rather than any disease.
- Regarding the Japanese study referenced as evidence for superior boosting from ARCT-154 compared to BNT162b2 – they should point out that this may be explained by the advantage of a heterologous combination over homologous. To truly assess this they would need to randomise the priming regimen and the boost.
- The authors claim that the AEs and local reactions were less frequent than licensed mRNA vaccines but they are disingenuous to reference a small phase I study, especially as their own vaccine was less well tolerated in the earlier phase. The mRNA vaccines have proved to be well tolerated in public health programmes and I don't see any justification for this claim.
- The duration of immune responses in animals and species are not defined so that paragraph is meaningless to the reader.
- Phase 3c data are not presented here and are the most relevant for evaluating this candidate vaccine in comparison to existing vaccine, but I can understand why the investigators want to publish now.
- Is this the first self-amplifying efficacy trial to be published? Perhaps so, but what about Gemcovac-19 which is a licensed vaccine against COVID-19 using alphavirus self-amplifying technology?
- If not the first, a key characteristic worth emphasising is the low dose compared to mRNA. The authors do not mention this advantage and I am not an expert in the costs of manufacturing so perhaps it is not the case that there are savings to be made with the lower dose.

Reviewer #3:

Remarks to the Author:

Summary

This study entitled "Safety and immunogenicity and efficacy of the self-amplifying mRNA ARCT-154 COVID-19 vaccine" investigates the safety, immunogenicity and efficacy of an integrated phase 1/2/3a/3b saRNA vaccine trial in Vietnam. The authors demonstrate that the ARCT-154 saRNA vaccine showed 95.3% efficacy against severe COVID-19 and associated disease.

Significantly, this is the first time an saRNA vaccine has been taken as far as a phase 3b efficacy trial and been shown to be efficacious against causing severe disease and associated death.

Wider Comments

While the paper is generally well written, the authors have tried to include too much information within this paper, which makes it difficult to read and means that some important data has been omitted. While licenced mRNA vaccines, mRNA-1273 and BNT162b2, used integrated phase 1/2/3 trials to accelerate their vaccines, they openly published the phase 1/2 data first, showing the full immunogenicity. Although the authors acknowledge that this is a limitation of this study, given the infancy of saRNA vaccines in the field of infectious disease, it is critical that these data are separated so that all information is transparent.

Szubert et al (<https://doi.org/10.1016/j.eclim.2022.101823>) demonstrated that an saRNA vaccine is capable of boosting pre-existing immune responses, yet no group have sufficiently demonstrated that saRNA can be used to prime an immune response in naïve individuals. Therefore, for the wider scientific field, it is important to understand the immunology behind what is happening after the first

dose. While saRNA clearly boosts immune responses (Szubert et al, 2023) and is efficacious, as shown by the authors of this current study, the lack of early study data and reported immunogenicity after one dose is a gaping hole in the data and would benefit from a more detailed immunogenicity paper prior to/in parallel with this efficacy paper.

While a previous version of the vaccine, ARCT-021, demonstrated an immune response after the first dose, this was not boosted by the second dose. In the paper currently being reviewed, the authors used their neutralisation assay to show that there is an immune response after one dose, which is boosted by a second dose. However, it is unclear if the participants of this trial were naïve to SARS-CoV-2 at baseline. The study would benefit from more antibody data in the form of ELISA or MSD assays.

Specific comments and suggestions:

1. Title: According to the CONSORT guidelines, the title should have the phrase "randomised trial" within it.
2. Reference 18 needs updating to "npj Vaccines (2022)7:161 ; <https://doi.org/10.1038/s41541-022-00590-x>"
3. Table 1 would benefit from some statistics to show that the demographics in the different phases of the trial are comparable to one another.
4. Table 2 would also benefit from some statistics to demonstrate that there is no significant difference between adverse events observed in the different phases of the trial.
5. Line 122 reads "rates systemic rates", which doesn't make sense.
6. The neutralisation assay used within the study is the only immunogenicity data given within this paper. This data would be better presented and easier to interpret in graphical form rather than a table.
7. The neutralisation data would be further strengthened by a live neutralisation assay and the inclusion of a comparison to sera from individuals who have received a licensed mRNA, as well as convalescent serum.
8. The authors need to be careful when using the term "seroconverting" when describing the immune response of participants to the vaccine. The recruited participants were not screened for SARS-CoV-2 S- or N-specific antibodies so it is unknown whether they have seroconverted or a low baseline immunity has been boosted by the first dose.
9. Figure 5 - the graph begins at D35. This does not allow the reader to make a judgement as to whether efficacy can be observed after one dose and is important as licensed mRNA vaccines have demonstrated a divergence from placebo from 12 days post prime onwards. This data is only partially reported within the supplementary figures (the placebo and the vaccinees have not been separated within the supplementary table) and should be within the graph.
10. Figure 5 - the numbers underneath the graph do not follow what is written within the text. For example, the text states that 43 severe cases of COVID-19 were detected between days 35-92, but the figures at the bottom add up to more than 43. Were the individuals counted twice to give a cumulative score within the graph? A table to breakdown the time of COVID-19 onset would be useful (as was used in DOI: 10.1056/NEJMoa2035389; Figure 3).
11. Table 4 - have the relevant statistics been applied to justify the comment that the rates of infection were similar in male and female participants?
12. There is no figure or table that breaks down the results of the experiments performed to record the variants responsible for the SARS-CoV-2 infections. The paper would benefit from this being graphically represented, at least in the supplementary figures.
13. On line 209, the authors make the statement that the efficacy estimates were similar in the 1-29 day time frame, yet this period has been omitted from the results and is not clearly evident in the data presented, so this statement should be removed as it cannot be proven with the data in the current format.
14. Authors should avoid using the use of geographical location terms such as "Wuhan-Hu-1" strain. "Wildtype" is a better term.
15. On line 222, the authors use the abbreviation "VE", which presumably represents "Vaccine

Efficacy" however, this is not stated anywhere within the paper, so needs defining.

16. The authors need to be clearer in the discussion that vaccine efficacy against the licenced mRNA vaccines was defined as "preventing COVID-19 illness, including severe disease", rather than "severe disease and death", which is the wider definition used used for the saRNA within this paper.

17. The authors postulate the difference in efficacy is lower with their vaccine due to the timing of the efficacy trial and their definition of COVID-19 disease, but it is not clear to the reader that, when compared to mRNA vaccines, their vaccine has an efficacy of 56.6% versus 95% (BNT162b2) and 94.1% (mRNA-1273). This needs to be more explicitly stated.

18. The statement on lines 225-227 "it is acknowledged that the effectiveness of COVID-19 vaccination increases with the severity of COVID-19 and decreases with time since vaccination" needs referencing.

19. The sentence between lines 230-234 beginning "In a population..." needs to be put into context of the vaccine being trialled. It is a valid statement, but it lacks context.

20. The sentence between lines 239-241 beginning "Although the trials were not identical in design so not directly comparable..." appears to contradict itself as the authors go on to compare licenced mRNA vaccines to ARCT-154.

21. The sentence beginning on line 247 "This study probably represents..." is misleading as the authors have not tested the population used within this study for S- or N- specific antibodies to SARS-CoV-2 at baseline, so it cannot be proven that this is a truly naïve population.

22. The authors need to comment that licenced mRNA vaccines are/have been updated to reflect the changes in circulating strains. These updated vaccines are being given to older/at risk populations in many countries, which is helping to boost the waning immunity within these populations. saRNA vaccines can, of course, be an additional tool.

23. The authors need to reference Szubert et al (2023) in line 258 who also demonstrated that a SARS-CoV-2 saRNA vaccine was capable of boosting antibodies and neutralisation activity in individuals with pre-existing immunity.

REVIEWER COMMENTS

Reviewer #1 (Remarks to the Author):

The authors conducted a comprehensive set of clinical trials evaluating the safety, immunogenicity and efficacy of ARCT-154, a VEEV replicon based self-amplifying RNA (saRNA) vaccine encoding the full-length spike protein of SARS-CoV-2 with D614G, as the primary series in a population mostly without prior SARS-CoV-2 infection. This is the saRNA-based COVID vaccine with the best reacto, immunogenicity and efficacy profiles and kudos to the authors for conducting a nice set of trials for the great info about this vaccine candidate. Given the extensive clinical and observational studies for BNT162b2 and mRNA-1273, however, accuracy needs to be maintained with comparison is described in the manuscript:

1. Line 230-234: Given the trial is measuring VE against Delta, it's not fair to reference a paper describing VE of the two commercial COVID vaccines against Omicron in order to show case the ARCT-154. There is no VE against Omicron/BA.1 or any durability data from ARCT-154 in this manuscript, so it's critical to stay in course when comparing this vaccine with the others.

On reflection, we do acknowledge that reference to efficacy results of other COVID-19 vaccines against Delta variant is a more appropriate comparison than to the later Omicron subvariants family, so we have replaced reference 5 with a more appropriate citation: Bernal JL, Andrews N, Gower C, et al. Effectiveness of Covid-19 Vaccines against the B.1.617.2 (Delta) Variant. N Engl J Med 2021; 385:585–94. DOI: 10.1056/NEJMoa2108891

We have also removed the following sentence (and the original reference 21) as this is Omicron-related (line 228): "In a population who mainly received mRNA vaccine, effectiveness of booster vaccination against severe COVID-19 decreased from 80·9% (95% CI: 76·9–84·2) within 7–30 days after booster vaccination to 63·4% (50·5–72·9) at 91–120 days post-vaccination, with little effect against mild COVID-19."

2. In numerous observational studies, VE (effectiveness) of the primary series of the two commercial COVID vaccine against Delta infection or symptomatic infection has been shown to be above 70%. It's worth to reference and discuss those studies in the Discussion. This will not only help the readers to understand the differences of COVID shots but also provide valuable insights to the sponsor of this study on prospect of this candidate product.

We have noted in the text that "The effectiveness of a completed primary vaccination series of authorized COVID-19 vaccines against infection by the Delta variant has been shown to range between 46% and 91%, and against severe COVID-19 disease due to Delta variant from 47% to 97%." (lines 222-225) and that "the lower VE of the ARCT-154 vaccine we observed may be a consequence of the definition of COVID-19 disease used in the trial, being based on presence of a single symptom in combination with positive RT-PCR." (lines 226-231). Clinical efficacy studies of other vaccines defined COVID-19 disease as a positive RT-PCR in combination with at least two systemic symptoms or at least one of the respiratory signs or symptoms. Our "easier to meet" definition would have allowed the inclusion of mild and marginally symptomatic cases in the analysis; it is acknowledged that the effectiveness of COVID-19 vaccination increases with the severity of COVID-19 and decreases with time since vaccination.

Reviewer #2 (Remarks to the Author):

- The authors present safety and per protocol efficacy data from Phases 1, 2, 3a and 3b of an ongoing study evaluating a candidate saRNA vaccine which provides up to three months follow-up, so too short a duration to assess whether these results are noteworthy in the context of licensed COVID19 vaccines.

We would note that efficacy estimates for the mRNA licensed vaccines were performed over periods of 112 days after the first dose for BNT162b2 (Polack FP, et al. Safety and efficacy of the BNT162b2 mRNA Covid-19 vaccine. NEJM 2020 31;383(27):2603–15) and 120 days for mRNA-1273 (Baden LR, et al. Efficacy and safety of the mRNA-1273 SARS-CoV-2 vaccine. NEJM 2021;384(5):403–16), so our surveillance for three months after the second dose (so 112 days from first dose) is consistent with those studies. Note that the study was conducted during the implementation of COVID-19 vaccination in Vietnam by national regulatory authorities and an extended follow-up period for placebo recipients was ethically and programmatically unacceptable. We agree that the assessment of longer-term efficacy, as with the licensed COVID-19 vaccines, would require post-registration effectiveness studies.

- Nonetheless, an impressive preliminary clinical trial dataset with results that are worthy of publication subject to clarification.
- I am not a virologist, but the assays used to assess seroconversion, defined as 4-fold change in GMT from baseline, are both validated and were conducted in independent laboratories which I imagine are subject to inspection of their quality systems.
- Assessment of clinical endpoints follows FDA guidance and was subject to adjudication by independent clinical experts who were blind to allocation.
- Consort and Tables 2 and 4 – I found it difficult to follow the numbers. What exactly do the authors mean by ‘exposed’ in the consort? Numbers do not match with second dose in Table 1.

Exposed is a general clinical trial expression to indicate that a participant received at least one dose of their assigned study intervention, i.e., an injection of vaccine or placebo. For clarity we have changed the term to “treated” (line 86). In Figure 1 we have substituted “exposed” with “vaccinated” or “injected” in ARCT-154 and Placebo groups, respectively.

We have corrected Figure 1 following further quality control checks of the data, and now we see in Figure 1 that 733 and 7869 received a second ARCT-154 vaccination in Phase 1/2/3a and Phase 3b, respectively, together with 245 and 78931 placebo injections. These numbers correlate with the numbers in Table 1 for Phases 1, 2 and 3a combined and Phase 3b.

2,295 (14%) participants withdrew from the phase 3b study and 79 (8%) from phases 1/2/3a; this is given as the main reason for ‘study treatment discontinuation’ but makes no sense to me when viewed alongside the numbers in table 1 for compliance with second dose. So what do they mean by ‘study treatment discontinuation’? Were they lost after the second dose, but before the 92 day cut-off? If so, readers need to know as this amounts to 14% which seems high to me as a loss from the main analyses and a point to be mentioned in the discussion with respect to limitations. It seems to me they have included everyone who entered the mITT, but what I would like to know is how many participants do they know the clinical status of

at day 92? It is this number and proportion that gives me confidence in the results – more so than person years and rates which can mask loss of those more vulnerable to severe disease and therefore endpoints.

The authors revised Figure 1 to indicate the number of randomized subjects per group (ARCT-154 and placebo) and cohort (Phase 1/2/3a and Phase 3b), number of subjects received Dose 1 and Dose 2 of the vaccine or placebo, number of subjects completed Day 29, Day 57 and Day 92 visits. In addition, the reasons for exclusion from main analysis populations (mITT, IAS, PPS-efficacy and SAS) and reasons for exclusions from each population are updated. We hope it addresses the comment and apologies for some confusion.

Table 1 represents the total enrolled population; Table 2 represents the Safety population, i.e., participants who received at least one dose of vaccine or placebo and provided some post-vaccination safety data.

Table 4 represents efficacy results for the modified ITT (mITT) population used for the primary efficacy analysis. mITT population includes all participants who received all protocol-required doses of study vaccine (ARCT-154 or placebo) up to Day 92, and who have no evidence of SARS CoV 2 infection on Day 1 or up to 7 days after the 2nd study vaccination. For efficacy analysis from Day 1 to Day 92, the ITT population with no evidence of infection prior to vaccination was used, which includes all subjects who received at least one dose of the study vaccine, had negative results for anti-nucleocapsid antibody test on Day 1, and provided some efficacy data post-vaccination.

- The safety profile presented is similar to licensed mRNA vaccine with more reactogenicity than recombinant protein vaccines. The candidate vaccine was sufficiently well tolerated for almost all participants to return and receive the second vaccine, and reactogenicity appeared to diminish as the size of the trial and age of participant increased.
- Participants had to be naïve to vaccines, and report no history of COVID-19 or a positive test. Nonetheless they could have had asymptomatic natural infection, or a symptomatic infection that they chose not to report. From Table 4 it seems anti-N was assessed for all participants to generate the dataset with no evidence of infection at baseline. and remarkably few had these antibodies if the heading for so include the n(%) with anti-nucleocapsid antibodies and whether this made a difference to inclusion in the analysis sets eg were they excluded from the mITT dataset for 3b?

It is correct that due to quarantine and social isolation measures implemented in Vietnam, the majority of the study population remained SARS-CoV-2 naïve. Anti-N antibody testing was performed at baseline in all study participants and only anti-N negative individuals with no history of COVID-19 disease or COVID-19 vaccination were eligible for the mITT population included in the primary efficacy analyses. We revised the manuscript, including the information of anti-N antibody testing in the Materials and Methods (lines 345-6 and 416).

- The immunogenicity results in table 3 support the clinical efficacy in table 4 with high rates of seroconversion after 2 doses and a strong effect for protection against severe COVID19 respectively. This interpretation is subject to my caveats above about the numbers lost to follow-up.
- Good to see the clinical effectiveness is similarly high in older age-groups and those with risk factors.
- The authors note that the strength of this candidate is in protection against severe disease rather than any disease.

- Regarding the Japanese study referenced as evidence for superior boosting from ARCT-154 compared to BNT162b2 – they should point out that this may be explained by the advantage of a heterologous combination over homologous. To truly assess this they would need to randomise the priming regimen and the boost.

We agree that an increment of the differences alluded to in the booster paper may be attributed to heterologous vs homologous boosting. Ideally, an assessment would be done using different priming vaccines, which is something we have investigated in the Phase 3c study. Nonetheless, the current global epidemiology of persons having immunity due to either natural infection with SARS-CoV-2, prior vaccination or hybrid immunity due to both infection and vaccination means that booster vaccination is the most probable use for any new vaccines such as sa-RNA. Thus, data showing a higher response to sa-RNA is important to inform the real-world use of the new vaccine. This will be further complicated by the use of new formulations based on the S-protein of later variants, e.g. the current requirement to use XBB.1.5.

- The authors claim that the AEs and local reactions were less frequent than licensed mRNA vaccines but they are disingenuous to reference a small phase I study, especially as their own vaccine was less well tolerated in the earlier phase. The mRNA vaccines have proved to be well tolerated in public health programmes and I don't see any justification for this claim.

We have presented the totality of reactogenicity data presented in this study, including pooled Phase 1/2/3a results (N = 1001) and Phase 3b (N = 16,100). To our knowledge, this is the largest safety set generated for sa-RNA vaccines so far. Indeed, the frequency of solicited reactions were higher in Phase 1/2/3a part of the study than in Phase 3b, which may reflect the differences in population (more elderly subjects in Phase 3b and better tolerability of the vaccines in the elderly), as well as differences in the data collection approach (e-diary use in Phase 1/2/3a vs mainly paper diary use in Phase 3b). However, the rates of solicited AEs in both cohorts were lower than published data for conventional mRNA vaccines. In the Discussion section, we have specifically indicated limitations associated with inter-study comparisons for safety and reactogenicity results (line 244).

- The duration of immune responses in animals and species are not defined so that paragraph is meaningless to the reader.

We agree and we have included “Vaccine effectiveness against the ancestral SARS-CoV-2 of two doses of either of the two main licensed mRNA vaccines, BNT162b2 and mRNA-1273, has been calculated to decline to 50.8% (95% CI: 19.7, 69.8) by 7 months after the second dose.” (lines 65-68) to indicate the duration of immunity of the licensed mRNA vaccines.

- Phase 3c data are not presented here and are the most relevant for evaluating this candidate vaccine in comparison to existing vaccine, but I can understand why the investigators want to publish now.

We plan to present the results of Phase 3c sub-study in a separate publication.

- Is this the first self-amplifying efficacy trial to be published? Perhaps so, but what about Gemcovac-19 which is a licensed vaccine against COVID-19 using alphavirus self-amplifying technology?

We were not able to find the results of. The safety and immunogenicity results of a Phase 3 efficacy study of Gemcovac in comparison with Covishield in 4000 participants are included in the Summary of Product Characteristics which can be accessed at

https://cdsco.gov.in/opencms/resources/UploadCDSCOWeb/2018/UploadSmPC/4.%20GEMCOVAC%20mRNA%20vaccine%20of%20Genova_SmPC,%20factsheet%20&%20PI.pdf but efficacy data was not included and we have been unable to locate a publication of the efficacy results.

- If not the first, a key characteristic worth emphasising is the low dose compared to mRNA. The authors do not mention this advantage and I am not an expert in the costs of manufacturing so perhaps it is not the case that there are savings to be made with the lower dose.

We have added mention of this potential advantage for sa-RNA to the Discussion (lines 284-288).

Reviewer #3 (Remarks to the Author):

Summary

This study entitled “Safety and immunogenicity and efficacy of the self-amplifying mRNA ARCT-154 COVID-19 vaccine” investigates the safety, immunogenicity and efficacy of an integrated phase 1/2/3a/3b saRNA vaccine trial in Vietnam. The authors demonstrate that the ARCT-154 saRNA vaccine showed 95.3% efficacy against severe COVID-19 and associated disease.

Significantly, this is the first time an saRNA vaccine has been taken as far as a phase 3b efficacy trial and been shown to be efficacious against causing severe disease and associated death.

Wider Comments

While the paper is generally well written, the authors have tried to include too much information within this paper, which makes it difficult to read and means that some important data has been omitted. While licenced mRNA vaccines, mRNA-1273 and BNT162b2, used integrated phase 1/2/3 trials to accelerate their vaccines, they openly published the phase 1/2 data first, showing the full immunogenicity. Although the authors acknowledge that this is a limitation of this study, given the infancy of saRNA vaccines in the field of infectious disease, it is critical that these data are separated so that all information is transparent.

The manuscript contains data from an extensive Phase 1/2/3a/3b study which was an approved format for clinical testing of new COVID-19 vaccines during the pandemic. We agree that the manuscript format does not allow the presentation of all the critical study results, but for a vaccine we believe the most important data are those that show the effectiveness of the vaccine to prevent the targeted disease as an efficacy calculation, which was the primary objective of the study and this paper. Efficacy studies of the two mRNA vaccines referred to were also published (Polack FP, et al. Safety and efficacy of the BNT162b2 mRNA Covid-19 vaccine. NEJM 2020 31;383(27):2603–15 and Baden LR, et al. Efficacy and safety of the mRNA-1273 SARS-CoV-2 vaccine. NEJM 2021;384(5):403–16). Additional data collected during our study, both the safety and reactogenicity data, and the summary of immunogenicity data from Phases 1/2/3a will be presented in more detail in separate manuscripts which are currently being prepared to allow adequate space to discuss these data within the limitations of scientific manuscripts/papers.

Szubert et al (<https://doi.org/10.1016/j.eclinm.2022.101823>) demonstrated that an saRNA vaccine is capable of boosting pre-existing immune responses, yet no group have sufficiently demonstrated that saRNA can be used to prime an immune response in naïve individuals. Therefore, for the wider scientific field, it is important to understand the immunology behind what is happening after the first dose. While saRNA clearly boosts immune responses (Szubert et al, 2023) and is efficacious, as shown by the authors of this current study, the lack of early study data and reported immunogenicity after one dose is a gaping hole in the data and would benefit from a more detailed immunogenicity paper prior to/in parallel with this efficacy paper.

As discussed above, we plan to present the immunogenicity of the vaccine, including kinetics of immune response after the first and second doses and durability of immune response after the primary vaccination

series vs. authorized AZ COVID-19 vaccine in a separate manuscript. The immunogenicity data after booster dose of the study vaccine from the pivotal Phase 3 study in Japan are in a paper which is In Press having been accepted for publication by the Lancet Infectious Diseases journal. We also plan to publish antibody persistence data after booster vaccination in comparison with the licensed mRNA vaccines in another standalone paper.

While a previous version of the vaccine, ARCT-021, demonstrated an immune response after the first dose, this was not boosted by the second dose. In the paper currently being reviewed, the authors used their neutralisation assay to show that there is an immune response after one dose, which is boosted by a second dose. However, it is unclear if the participants of this trial were naïve to SARS-CoV-2 at baseline. The study would benefit from more antibody data in the form of ELISA or MSD assays.

We confirm that there were significant improvements made in the development of ARCT-154 vaccine compared with ARCT-021, including the use of Spike glycoprotein of the ancestral SARS-CoV-2 strain with a D614G mutation, proline substitutions resulting in the S glycoprotein being expressed in the prefusion conformation and furin cleavage site modification to improve the stability (lines 81-2). Additional changes include optimization of replicon and modification of vaccine impurity profile. All these changes were associated with increased immunogenicity and an improved tolerability profile compared with the parent vaccine.

Authors confirmed that immunogenicity data presented for SARS-CoV-2 naïve population, defined as subjects without history of COVID-19 disease or COVID-19 vaccination and without anti-nucleocapsid antibodies at the time of recruitment (lines 337-8 and 408).

Specific comments and suggestions:

1. Title: According to the CONSORT guidelines, the title should have the phrase “randomised trial” within it.

We have modified the title accordingly, to read “pooled phase 1, 2, 3a and 3b randomized, controlled trials”

2. Reference 18 needs updating to “npj Vaccines (2022)7:161 ; <https://doi.org/10.1038/s41541-022-00590-x>”;

We have updated what is now Reference 19.

3. Table 1 would benefit from some statistics to show that the demographics in the different phases of the trial are comparable to one another.

No statistical comparisons were made between the four independent cohorts – the results all lie within the same range.

4. Table 2 would also benefit from some statistics to demonstrate that there is no significant difference between adverse events observed in the different phases of the trial.

A statistical comparison between the different phases would not be appropriate due to the differences in populations recruited for each phase.

5. Line 122 reads “rates systemic rates”, which doesn’t make sense.

This is a typo and should read “rate of systemic adverse events” and has been corrected (lines 124-5)

6. The neutralisation assay used within the study is the only immunogenicity data given within this paper. This data would be better presented and easier to interpret in graphical form rather than a table.

As already noted, the full immunogenicity data set will be presented separately so we believe a Table with geometric mean concentrations, seroconversion rates and geometric mean-fold rises in neutralizing antibodies which are considered the gold standard to measure immune responses to COVID-19 vaccines is appropriate for this paper that is primarily focused on efficacy. The neutralizing antibody response was measured using two methods which essentially confirm each other’s results.

7. The neutralisation data would be further strengthened by a live neutralisation assay and the inclusion of a comparison to sera from individuals who have received a licence mRNA, as well as convalescent serum.

Again, we would emphasize that the primary objective of this paper was the efficacy in which the comparison was with placebo, not other COVID -19 vaccines. Such a comparison will be made against licensed COVID-19 vaccines (Comirnaty and Vaxzevria) or convalescent plasma in ongoing and future papers.

8. The authors need to be careful when using the term “seroconverting” when describing the immune response of participants to the vaccine. The recruited participants were not screened for SARS-CoV-2 S- or N-specific antibodies so it is unknown whether they have seroconverted or a low baseline immunity has been boosted by the first dose.

All study participants, included in the primary immunogenicity analysis were SARS-CoV-2 naïve, as defined by the result of anti-nucleocapsid antibody test on Day 1.

We are using the SCR/SRR definition proposed in the current guidelines for COVID-19 vaccine development. The term seroconvert is specifically defined in the Statistics section and indicates a four-fold or greater increase in antibody concentration from baseline to postvaccination, irrespective of the serostatus of the participant before vaccination.

9. Figure 5 - the graph begins at D35. This does not allow the reader to make a judgement as to whether efficacy can be observed after one dose and is important as licenced mRNA vaccines have demonstrated a divergence from placebo from 12 days post prime onwards. This data is only partially reported within the supplementary figures (the placebo and the vaccinees have not been separated within the supplementary table) and should be within the graph.

In Supplementary figure 2 we present the cumulative distribution curves from Day 1 that confirm the observation that the main divergence from placebo occurred after 7 days post-Dose 2

10. Figure 5 – the numbers underneath the graph do not follow what is written within the text. For example, the text states that 43 severe cases of COVID-19 were detected between days 35-92, but the figures at the bottom add up to more than 43. Were the individuals counted twice to give a cumulative score within the graph? A table to breakdown the time of COVID-19 onset would be useful (as was used in DOI: 10.1056/NEJMoa2035389; Figure 3).

The figures, as noted in the title, show COVID-19 of any severity in the upper panel, and severe COVID-19 in the lower panel. We have updated the Figure to include all the cumulative numbers of severe cases up to Day 92 showing the 43 cases (41 placebo, 2 vaccine).

11. Table 4 – have the relevant statistics been applied to justify the comment that the rates of infection were similar in male and female participants?

No pre-defined comparisons were done for subgroup analyses

12. There is no figure or table that breaks down the results of the experiments performed to record the variants responsible for the SARS-CoV-2 infections. The paper would benefit from this being graphically represented, at least in the supplementary figures.

The overwhelming majority (477 of 537 [89%]) of cases of virologically-confirmed COVID-19 in which the strain was identified were Delta. We have included this information in a new Supplementary table 5.

13. On line 209, the authors make the statement that the efficacy estimates were similar in the 1-29 day time frame, yet this period has been omitted from the results and is not clearly evident in the data presented, so this statement should be removed as it cannot be proven with the data in the current format.

We have removed this.

14. Authors should avoid using the use of geographical location terms such as “Wuhan-Hu-1” strain. “Wildtype” is a better term.

The term Wuhan-Hu1 strain is used to indicate the genetic sequence used for manufacturing of ARCT-021 vaccine and a test strain utilized in the neutralization assay. We believe that indication of an exact strain used provides more relevant information to readers compared with ‘wildtype strain’ which actually could apply to all circulating strains of SARS-CoV-2.

15. On line 222, the authors use the abbreviation “VE”, which presumably represents “Vaccine Efficacy” however, this is not stated anywhere within the paper, so needs defining.

We have defined VE as vaccine efficacy (line 169).

16. The authors need to be clearer in the discussion that vaccine efficacy against the licenced mRNA vaccines was defined as “preventing COVID-19 illness, including severe disease”, rather than “severe disease and death”, which is the wider definition used used for the saRNA within this paper.

We have included “against COVID-19 illness, including severe disease,” in the text (line 218).

17. The authors postulate the difference in efficacy is lower with their vaccine due to the timing of the efficacy trial and their definition of COVID-19 disease, but it is not clear to the reader that, when compared to mRNA vaccines, their vaccine has an efficacy of 56.6% versus 95% (BNT162b2) and 94.1% (mRNA-1273). This needs to be more explicitly stated.

First, we should indicate that this lower VE of the ARCT-154 vaccine was against Delta variant, and we have already put in text that “The effectiveness of a completed primary vaccination series of authorized COVID-19

vaccines against infection by the Delta variant has been shown to range between 46% and 91%, and from 47% to 97% against severe COVID-19 disease due to Delta variant.” (lines 222-5)

The lower VE we observed may also be a consequence of the definition of COVID-19 disease used in the trial, being based on the presence of a single symptom in combination with a positive RT-PCR. This allows the inclusion of a significant number of mild and marginally symptomatic cases in the analysis, and it is acknowledged that the effectiveness of COVID-19 vaccination increases with the severity of COVID-19. For example, the effectiveness of a 3-dose vaccination series of BNT162b2 against asymptomatic and symptomatic COVID-19 disease during Omicron BA.2 outbreak was different - 41.4% (95% CI: 23.2–55.2; $p = 0.0001$) and 50.9% (95% CI 31.0–65.0; $p < 0.0001$), respectively. Furthermore, the effectiveness of a 3-dose series of BNT162b2 vaccine against mild, moderate and severe COVID-19, caused by the Omicron variant, within 31 to 60 days after the last vaccination, was 7.9% (95% CI: 2.3–13.1), 49.2% (95% CI: 46.8–51.4) and 76.4% (95% CI: 72.4–79.8), respectively.²² It was notable that ARCT-154 was much more efficacious against severe COVID-19 than disease of any severity. As such, the case definition used for primary efficacy analysis has a significant impact on the efficacy point estimate. Clinical studies with other vaccines used a ‘more symptomatic’ definition of COVID-19 disease (a positive RT-PCR in combination with at least two systemic symptoms or at least one the respiratory signs or symptom). (Lines 235-239)

18. The statement on lines 225-227 “it is acknowledged that the effectiveness of COVID-19 vaccination increases with the severity of COVID-19 and decreases with time since vaccination” needs referencing.

Addressed in the response above.

19. The sentence between lines 230-234 beginning “In a population...” needs to be put into context of the vaccine being trialled. It is a valid statement, but it lacks context.

We have deleted this phrase and replaced it as above.

20. The sentence between lines 239-241 beginning “Although the trials were not identical in design so not directly comparable...” appears to contradict itself as the authors go on to compare licenced mRNA vaccines to ARCT-154.

Indeed, we compared the results across the trials but the methodologies employed and adverse events solicited in clinical trials are sufficiently similar to allow such a general comparison.

21. The sentence beginning on line 247 “This study probably represents...” is misleading as the authors have not tested the population used within this study for S- or N- specific antibodies to SARS-CoV-2 at baseline, so it cannot be proven that this is a truly naïve population.

Anti-N seropositivity was tested at baseline in all participants and the primary efficacy analysis and presented immunogenicity analyses only include SARS-CoV-2 naïve subjects based on medical history and serology for N-specific antibodies. This is now indicated (lines 337-8 and 408).

22. The authors need to comment that licenced mRNA vaccines are/have been updated to reflect the changes in circulating strains. These updated vaccines are being given to older/at risk populations in many countries,

which is helping to boost the waning immunity within these populations. saRNA vaccines can, of course, be an additional tool.

We have added a comment to this effect (lines 283-2887).

23. The authors need to reference Szubert et al (2023) in line 258 who also demonstrated that a SARS-CoV-2 saRNA vaccine was capable of boosting antibodies and neutralisation activity in individuals with pre-existing immunity.

We were unaware of this manuscript at the time of writing but are grateful to the reviewer for pointing it out and have now added it to our manuscript (lines 272-275).

Reviewers' Comments:

Reviewer #3:

Remarks to the Author:

Responses are in the attached document

Reviewer #3 (Remarks to the Author):

Summary

This study entitled “Safety and immunogenicity and efficacy of the self-amplifying mRNA ARCT-154 COVID-19 vaccine” investigates the safety, immunogenicity and efficacy of an integrated phase 1/2/3a/3b saRNA vaccine trial in Vietnam. The authors demonstrate that the ARCT-154 saRNA vaccine showed 95.3% efficacy against severe COVID-19 and associated disease.

Significantly, this is the first time an saRNA vaccine has been taken as far as a phase 3b efficacy trial and been shown to be efficacious against causing severe disease and associated death.

Wider Comments

While the paper is generally well written, the authors have tried to include too much information within this paper, which makes it difficult to read and means that some important data has been omitted. While licenced mRNA vaccines, mRNA-1273 and BNT162b2, used integrated phase 1/2/3 trials to accelerate their vaccines, they openly published the phase 1/2 data first, showing the full immunogenicity. Although the authors acknowledge that this is a limitation of this study, given the infancy of saRNA vaccines in the field of infectious disease, it is critical that these data are separated so that all information is transparent.

The manuscript contains data from an extensive Phase 1/2/3a/3b study which was an approved format for clinical testing of new COVID-19 vaccines during the pandemic. We agree that the manuscript format does not allow the presentation of all the critical study results, but for a vaccine we believe the most important data are those that show the effectiveness of the vaccine to prevent the targeted disease as an efficacy calculation, which was the primary objective of the study and this paper. Efficacy studies of the two mRNA vaccines referred to were also published (Polack FP, et al. Safety and efficacy of the BNT162b2 mRNA Covid-19 vaccine. NEJM 2020 31;383(27):2603–15 and Baden LR, et al. Efficacy and safety of the mRNA-1273 SARS-CoV-2 vaccine. NEJM 2021;384(5):403–16). Additional data collected during our study, both the safety and reactogenicity data, and the summary of immunogenicity data from Phases 1/2/3a will be presented in more detail in separate manuscripts which are currently being prepared to allow adequate space to discuss these data within the limitations of scientific manuscripts/papers.

It was acknowledged by this reviewer in their initial response that integrated phase 1/2/3 trials were used to accelerate vaccines during the pandemic. It is also agreed that the key data for a vaccine study is endpoint efficacy. However, given the importance and potential of saRNA vaccines in future pandemics, it is critical that the immunogenicity papers are published. If this early data is ready to be made public, it would be nice to see them published in conjunction with this efficacy paper as there is a risk is that this data will not be published in the future (despite the best wishes of the authors).

Both the mRNA-1273 (<https://www.nejm.org/doi/full/10.1056/NEJMoa2022483>) and BNT162b2 (<https://www.nature.com/articles/s41586-020-2639-4>) vaccines published early safety and immunogenicity data before their full efficacy data, so this is not unusual. In addition, their work was published during 2020 at

the height of the pandemic. Given the infancy of saRNA vaccines, this reviewer needs assurances beyond the word of the authors that the preliminary data will be published.

Szubert et al (<https://doi.org/10.1016/j.eclinm.2022.101823>) demonstrated that an saRNA vaccine is capable of boosting pre-existing immune responses, yet no group have sufficiently demonstrated that saRNA can be used to prime an immune response in naïve individuals. Therefore, for the wider scientific field, it is important to understand the immunology behind what is happening after the first dose. While saRNA clearly boosts immune responses (Szubert et al, 2023) and is efficacious, as shown by the authors of this current study, the lack of early study data and reported immunogenicity after one dose is a gaping hole in the data and would benefit from a more detailed immunogenicity paper prior to/in parallel with this efficacy paper.

As discussed above, we plan to present the immunogenicity of the vaccine, including kinetics of immune response after the first and second doses and durability of immune response after the primary vaccination series vs. authorized AZ COVID-19 vaccine in a separate manuscript. The immunogenicity data after booster dose of the study vaccine from the pivotal Phase 3 study in Japan are in a paper which is In Press having been accepted for publication by the Lancet Infectious Diseases journal. We also plan to publish antibody persistence data after booster vaccination in comparison with the licensed mRNA vaccines in another standalone paper.

The future publications are exciting and should be in addition to already published data (see above comment).

While a previous version of the vaccine, ARCT-021, demonstrated an immune response after the first dose, this was not boosted by the second dose. In the paper currently being reviewed, the authors used their neutralisation assay to show that there is an immune response after one dose, which is boosted by a second dose. However, it is unclear if the participants of this trial were naïve to SARS-CoV-2 at baseline. The study would benefit from more antibody data in the form of ELISA or MSD assays.

We confirm that there were significant improvements made in the development of ARCT-154 vaccine compared with ARCT-021, including the use of Spike glycoprotein of the ancestral SARS-CoV-2 strain with a D614G mutation, proline substitutions resulting in the S glycoprotein being expressed in the prefusion conformation and furin cleavage site modification to improve the stability (lines 81-2). Additional changes include optimization of replicon and modification of vaccine impurity profile. All these changes were associated with increased immunogenicity and an improved tolerability profile compared with the parent vaccine.

Please include all this information within the main body of the introduction.

Authors confirmed that immunogenicity data presented for SARS-CoV-2 naïve population, defined as subjects without history of COVID-19 disease or COVID-19 vaccination and without anti-nucleocapsid antibodies at the time of recruitment (lines 337-8 and 408).

Please see comments in point 21, below.

Specific comments and suggestions:

1. Title: According to the CONSORT guidelines, the title should have the phrase “randomised trial” within it.

We have modified the title accordingly, to read “pooled phase 1, 2, 3a and 3b randomized, controlled trials”

Thank you

2. Reference 18 needs updating to “npj Vaccines (2022)7:161 ; <https://doi.org/10.1038/s41541-022-00590-x>”;

We have updated what is now Reference 19.

Thank you

3. Table 1 would benefit from some statistics to show that the demographics in the different phases of the trial are comparable to one another.

No statistical comparisons were made between the four independent cohorts – the results all lie within the same range.

Noted

4. Table 2 would also benefit from some statistics to demonstrate that there is no significant difference between adverse events observed in the different phases of the trial.

A statistical comparison between the different phases would not be appropriate due to the differences in populations recruited for each phase.

Thank you for clarifying. Please can a note of this be made within the text/table so it is clear to the readers.

5. Line 122 reads “rates systemic rates”, which doesn’t make sense.

This is a typo and should read “rate of systemic adverse events” and has been corrected (lines 124-5).

Thank you for correcting this.

6. The neutralisation assay used within the study is the only immunogenicity data given within this paper. This data would be better presented and easier to interpret in graphical form rather than a table.

As already noted, the full immunogenicity data set will be presented separately so we believe a Table with geometric mean concentrations, seroconversion rates and geometric mean-fold rises in neutralizing antibodies which are considered the gold standard to measure immune responses to COVID-19 vaccines is appropriate for this paper that is primarily focused on efficacy. The neutralizing antibody response was measured using two methods which essentially confirm each other’s results.

Thank you for the clarification. I agree that neutralizing antibodies are widely considered to be the gold standard for the measurement of immune responses, which is why I believe a graphical representation of this data will be easier for the readers to interpret and therefore should be included (even if it’s in the supplementary information). As this data is a combination of phases 1-3, it is important to clearly show this information within the body of the paper, even if it will be presented (presumably in greater detail) in a full immunogenicity data paper.

7. The neutralisation data would be further strengthened by a live neutralisation assay and the inclusion of a comparison to sera from individuals who have received a licence mRNA, as well as convalescent serum.

Again, we would emphasize that the primary objective of this paper was the efficacy in which the comparison was with placebo, not other COVID -19 vaccines. Such a comparison will be made against licensed COVID-19 vaccines (Comirnaty and Vaxzevria) or convalescent plasma in ongoing and future papers.

I agree that the primary focus of this paper is efficacy. Therefore, I agree that the live neutralization assay may be beyond the scope of this paper. However, as the immunogenicity data has yet to be published, and as mentioned above, a graphical representation of the neutralization data already performed should be included, alongside a comparison of sera from individuals who have received a licence mRNA, as well as convalescent serum.

8. The authors need to be careful when using the term “seroconverting” when describing the immune response of participants to the vaccine. The recruited participants were not screened for SARS-CoV-2 S- or N-specific antibodies so it is unknown whether they have seroconverted or a low baseline immunity has been boosted by the first dose.

All study participants, included in the primary immunogenicity analysis were SARS-CoV-2 naïve, as defined by the result of anti-nucleocapsid antibody test on Day 1.

See my response to point 21 below

We are using the SCR/SRR definition proposed in the current guidelines for COVID-19 vaccine development. The term seroconvert is specifically defined in the Statistics section and indicates a four-fold or greater increase in antibody concentration from baseline to postvaccination, irrespective of the serostatus of the participant before vaccination.

Noted

9. Figure 5 - the graph begins at D35. This does not allow the reader to make a judgement as to whether efficacy can be observed after one dose and is important as licenced mRNA vaccines have demonstrated a divergence from placebo from 12 days post prime onwards. This data is only partially reported within the supplementary figures (the placebo and the vaccinees have not been separated within the supplementary table) and should be within the graph.

In Supplementary figure 2 we present the cumulative distribution curves from Day 1 that confirm the observation that the main divergence from placebo occurred after 7 days post-Dose 2

Thank you for including this figure. It makes it much clearer to the reader that the main divergence occurs 7 days post 2nd dose. It is important to note in the discussion that this is different from mRNA vaccines, where the divergence began after 1 dose.

10. Figure 5 – the numbers underneath the graph do not follow what is written within the text. For example, the text states that 43 severe cases of COVID-19 were detected between days 35-92, but the figures at the bottom add up to more than 43. Were the individuals counted twice to give a cumulative score within the graph? A table to breakdown the time of COVID-19 onset would be useful (as was used in DOI: 10.1056/NEJMoa2035389; Figure 3).

The figures, as noted in the title, show COVID-19 of any severity in the upper panel, and severe COVID-19 in the lower panel. We have updated the Figure to include all the cumulative numbers of severe cases up to Day 92 showing the 43 cases (41 placebo, 2 vaccine).

It is understood that figure 5 represents COVID-19 of any severity in the upper panel, and severe COVID-19 in the lower panel. However, the confusion lay (and still lies) in where the figure of 43 comes from? If this is the total number (i.e., 41 in the placebo group and 2 in the vaccine group) then there should be an additional column in Figure 5 to denote this as I am not clear where the values come from (the total of the values in Figure 5B are 109 in the placebo group $[1+3+8+13+22+29+34=109]$ and 7 in the vaccinee group $[1+2+2+2]$). I suspect these are cumulative number, but this is not explicitly clear to me.

11. Table 4 – have the relevant statistics been applied to justify the comment that the rates of infection were similar in male and female participants?

No pre-defined comparisons were done for subgroup analyses

Noted

12. There is no figure or table that breaks down the results of the experiments performed to record the variants responsible for the SARS-CoV-2 infections. The paper would benefit from this being graphically represented, at least in the supplementary figures.

The overwhelming majority (477 of 537 [89%]) of cases of virologically-confirmed COVID-19 in which the strain was identified were Delta. We have included this information in a new Supplementary table 5.

Thank you for including this – it is much clearer for the reader.

13. On line 209, the authors make the statement that the efficacy estimates were similar in the 1-29 day time frame, yet this period has been omitted from the results and is not clearly evident in the data presented, so this statement should be removed as it cannot be proven with the data in the current format.

We have removed this.

Thank you

14. Authors should avoid using the use of geographical location terms such as “Wuhan-Hu-1” strain. “Wildtype” is a better term.

The term Wuhan-Hu1 strain is used to indicate the genetic sequence used for manufacturing of ARCT-021 vaccine and a test strain utilized in the neutralization assay. We believe that indication of an exact strain used provides more relevant information to readers compared with ‘wildtype strain’ which actually could apply to all circulating strains of SARS-CoV-2.

I would compromise here and request the authors denote the virus as SARS-CoV-2 WT (Wuhan-Hu-1). Furthermore, if the authors prefer to use the genetic sequence identity to indicate the exact strain, then use of “Delta” should be written as Delta (B.1.617.2) as should other variants.

15. On line 222, the authors use the abbreviation “VE”, which presumably represents “Vaccine Efficacy” however, this is not stated anywhere within the paper, so needs defining.

We have defined VE as vaccine efficacy (line 169).

Thank you for including this. However, the use of VE on line 222 of the original manuscript (now line 226) is the first mention of VE within the Discussion section and should, therefore, be redefined.

16. The authors need to be clearer in the discussion that vaccine efficacy against the licenced mRNA vaccines was defined as “preventing COVID-19 illness, including severe disease”, rather than “severe disease and death”, which is the wider definition used used for the saRNA within this paper.

We have included “against COVID-19 illness, including severe disease,” in the text (line 218).

Thank you

17. The authors postulate the difference in efficacy is lower with their vaccine due to the timing of the efficacy trial and their definition of COVID-19 disease, but it is not clear to the reader that, when compared to mRNA vaccines, their vaccine has an efficacy of 56.6% versus 95% (BNT162b2) and 94.1% (mRNA-1273). This needs to be more explicitly stated.

First, we should indicate that this lower VE of the ARCT-154 vaccine was against Delta variant, and we have already put in text that “The effectiveness of a completed primary vaccination series of authorized COVID-19 vaccines against infection by the Delta variant has been shown to range between 46% and 91%, and from 47% to 97% against severe COVID-19 disease due to Delta variant.” (lines 222-5)

The lower VE we observed may also be a consequence of the definition of COVID-19 disease used in the trial, being based on the presence of a single symptom in combination with a positive RT-PCR. This allows the inclusion of a significant number of mild and marginally symptomatic cases in the analysis, and it is acknowledged that the effectiveness of COVID-19 vaccination increases with the severity of COVID-19. For example, the effectiveness of a 3-dose vaccination series of BNT162b2 against asymptomatic and symptomatic COVID-19 disease during Omicron BA.2 outbreak was different - 41.4% (95% CI: 23.2–55.2; $p = 0.0001$) and 50.9% (95% CI 31.0–65.0; $p < 0.0001$), respectively. Furthermore, the effectiveness of a 3-dose series of BNT162b2 vaccine against mild, moderate and severe COVID-19, caused by the Omicron variant, within 31 to 60 days after the last vaccination, was 7.9% (95% CI: 2.3–13.1), 49.2% (95% CI: 46.8–51.4) and 76.4% (95% CI: 72.4–79.8), respectively.²² It was notable that ARCT-154 was much more efficacious against severe COVID-19 than disease of any severity. As such, the case definition used for primary efficacy analysis has a significant impact on the efficacy point estimate. Clinical studies with other vaccines used a ‘more symptomatic’ definition of COVID-19 disease (a positive RT-PCR in combination with at least two systemic symptoms or at least one the respiratory signs or symptom). (Lines 235-239)

Thank you for editing it – it is much clearer now.

18. The statement on lines 225-227 “it is acknowledged that the effectiveness of COVID-19 vaccination increases with the severity of COVID-19 and decreases with time since vaccination” needs referencing.

Addressed in the response above.

Thank you

19. The sentence between lines 230-234 beginning “In a population...” needs to be put into context of the vaccine being trialled. It is a valid statement, but it lacks context.

We have deleted this phrase and replaced it as above.

Thank you

20. The sentence between lines 239-241 beginning “Although the trials were not identical in design so not directly comparable...” appears to contradict itself as the authors go on to compare licenced mRNA vaccines to ARCT-154.

Indeed, we compared the results across the trials but the methodologies employed and adverse events solicited in clinical trials are sufficiently similar to allow such a general comparison.

In which case, would the authors consider rephrasing to “Although the trials were not identical in design, a general comparison determined that overall systemic AEs and local reactions were less frequent in recipients of ARCT-154 than licensed mRNA vaccines”.

21. The sentence beginning on line 247 “This study probably represents...” is misleading as the authors have not tested the population used within this study for S- or N- specific antibodies to SARS-CoV-2 at baseline, so it cannot be proven that this is a truly naïve population.

Anti-N seropositivity was tested at baseline in all participants and the primary efficacy analysis and presented immunogenicity analyses only include SARS-CoV-2 naïve subjects based on medical history and serology for N-specific antibodies. This is now indicated (lines 337-8 and 408).

Thank you for including this within the text – however, I would ideally like to see this in supplementary results and then the results referenced within the text. Having trawled through all the datasets, I cannot see any evidence that testing for N-antibodies was performed. Indeed, within the “exclusion criteria” in supplementary table 2, point 3, line 613 reads “nucleocapsid positive test is not exclusionary”, which seems to contradict the statements within the body of the paper.

22. The authors need to comment that licenced mRNA vaccines are/have been updated to reflect the changes in circulating strains. These updated vaccines are being given to older/at risk populations in many countries, which is helping to boost the waning immunity within these populations. saRNA vaccines can, of course, be an additional tool.

We have added a comment to this effect (lines 283-287).

Thank you

23. The authors need to reference Szubert et al (2023) in line 258 who also demonstrated that a SARS-CoV-2 saRNA vaccine was capable of boosting antibodies and neutralisation activity in individuals with pre-existing immunity.

We were unaware of this manuscript at the time of writing but are grateful to the reviewer for pointing it out and have now added it to our manuscript (lines 272-275).

Thank you

Reviewer #4:

Remarks to the Author:

Per the Editor's request, this reviewer focused rather narrowly on determining if Reviewer #2's comments were adequately addressed in the revision. Comments which were considered adequately resolved are not repeated in this review, and only items remaining to be addressed are highlighted here:

1. Reviewer #2 found it difficult to match numbers vaccinated across Tables 1, 2, 4, and the CONSORT diagram. The rebuttal cites changes in terminology and minor corrections of numbers in Figure 1 (actually, Figure 2). There is at least one remaining issue: the column N's used in Table 2 cannot be found on the CONSORT, and it is unclear why. Per the protocol, the Safety Set should have been used for Table 2, defined as the number vaccinated. The numbers vaccinated in Phases 1/2/3a/3b in the CONSORT differ from column N's in table 2. In addition, Table 1 would appear to be a summary among the Randomized Set with n=750 randomized to receive ARCT-154 in phase 1/2/3a (i.e., there is no "Enrolled Set" defined in the protocol), but Figure 2 shows 749 randomized into the ARCT-154 arm in phase 1/2/3a; I did not identify a reason for these minor differences. The analysis population, using terminology from the protocol, should be identified in the caption or footnotes for all tables.

2. Reviewer #2 did not find sufficient evidence to warrant the claim that solicited AEs were lower in the present study relative to those from external studies of alternative mRNA COVID-19 vaccines. The revised manuscript aims to strengthen the evidence supporting the claim, citing similarity of data collection/reporting methods. However, Reviewer #2's main concern has not been fully addressed, as a number of issues should be either accounted for in a supportive analysis to strengthen the claim, or instead cited as weaknesses of the comparison including potential cultural differences in reporting of subjective events such as headache and myalgia, as well as potentially varying mixtures of ages included in each study, which is generally understood to be related to rates of reactogenicity for COVID-19 vaccines.

Overall the manuscript appeared well-written, presenting noteworthy efficacy results for the novel self-amplifying mRNA vaccine candidate. A suggestion to the authors for the analysis following Day 92 is to take great care with how to present efficacy data that will presumably have two different risk periods for vaccinees (full duration for those initially vaccinated and post-D92 for controls who are switched over, e.g. no controls in the post-D92 period). Similar considerations will apply for safety (i.e., lumping together reactions to placebo doses may or may not be appropriate for those initially given placebo injections versus those given placebo injections at D92, after having previously received two vaccinations). This seems partially, but not fully addressed in the SAP shared for the present analysis.

Reviewer #3 (Remarks to the Author):

The authors' responses to Reviewer 3 are presented in green below, and to Reviewer 4 in Red.

Summary

This study entitled "Safety and immunogenicity and efficacy of the self-amplifying mRNA ARCT-154 COVID-19 vaccine" investigates the safety, immunogenicity and efficacy of an integrated phase 1/2/3a/3b saRNA vaccine trial in Vietnam. The authors demonstrate that the ARCT-154 saRNA vaccine showed 95.3% efficacy against severe COVID-19 and associated disease.

Significantly, this is the first time an saRNA vaccine has been taken as far as a phase 3b efficacy trial and been shown to be efficacious against causing severe disease and associated death.

Wider Comments

While the paper is generally well written, the authors have tried to include too much information within this paper, which makes it difficult to read and means that some important data has been omitted. While licenced mRNA vaccines, mRNA-1273 and BNT162b2, used integrated phase 1/2/3 trials to accelerate their vaccines, they openly published the phase 1/2 data first, showing the full immunogenicity. Although the authors acknowledge that this is a limitation of this study, given the infancy of saRNA vaccines in the field of infectious disease, it is critical that these data are separated so that all information is transparent.

The manuscript contains data from an extensive Phase 1/2/3a/3b study which was an approved format for clinical testing of new COVID-19 vaccines during the pandemic. We agree that the manuscript format does not allow the presentation of all the critical study results, but for a vaccine we believe the most important data are those that show the effectiveness of the vaccine to prevent the targeted disease as an efficacy calculation, which was the primary objective of the study and this paper. Efficacy studies of the two mRNA vaccines referred to were also published (Polack FP, et al. Safety and efficacy of the BNT162b2 mRNA Covid-19 vaccine. NEJM 2020 31;383(27):2603–15 and Baden LR, et al. Efficacy and safety of the mRNA-1273 SARS-CoV-2 vaccine. NEJM 2021;384(5):403–16). Additional data collected during our study, both the safety and reactogenicity data, and the summary of immunogenicity data from Phases 1/2/3a will be presented in more detail in separate manuscripts which are currently being prepared to allow adequate space to discuss these data within the limitations of scientific manuscripts/papers.

It was acknowledged by this reviewer in their initial response that integrated phase 1/2/3 trials were used to accelerate vaccines during the pandemic. It is also agreed that the key data for a vaccine study is endpoint efficacy. However, given the importance and potential of saRNA vaccines in future pandemics, it is critical that the immunogenicity papers are published. If this early data is ready to be

made public, it would be nice to see them published in conjunction with this efficacy paper as there is a risk is that this data will not be published in the future (despite the best wishes of the authors).

Both the mRNA-1273 (<https://www.nejm.org/doi/full/10.1056/NEJMoa2022483>) and BNT162b2 (<https://www.nature.com/articles/s41586-020-2639-4>) vaccines published early safety and immunogenicity data before their full efficacy data, so this is not unusual. In addition, their work was published during 2020 at the height of the pandemic. Given the infancy of saRNA vaccines, this reviewer needs assurances beyond the word of the authors that the preliminary data will be published.

Szubert et al (<https://doi.org/10.1016/j.eclinm.2022.101823>) demonstrated that an saRNA vaccine is capable of boosting pre-existing immune responses, yet no group have sufficiently demonstrated that saRNA can be used to prime an immune response in naïve individuals. Therefore, for the wider scientific field, it is important to understand the immunology behind what is happening after the first dose. While saRNA clearly boosts immune responses (Szubert et al, 2023) and is efficacious, as shown by the authors of this current study, the lack of early study data and reported immunogenicity after one dose is a gaping hole in the data and would benefit from a more detailed immunogenicity paper prior to/in parallel with this efficacy paper.

As discussed above, we plan to present the immunogenicity of the vaccine, including kinetics of immune response after the first and second doses and durability of immune response after the primary vaccination series vs. authorized AstraZeneca COVID-19 vaccine in a separate manuscript. The immunogenicity data after booster dose of the study vaccine from the pivotal Phase 3 study in Japan have been published by the Lancet Infectious Diseases journal [1]. We also published antibody persistence data after booster vaccination in comparison with the licensed mRNA vaccines in another standalone paper [2].

[1] Oda Y, Kumagai Y, Kanai M, et al. Immunogenicity and safety of a booster dose of a self-amplifying RNA COVID-19 vaccine (ARCT-154) versus BNT162b2 mRNA COVID-19 vaccine: a double-blind, multicentre, randomised, controlled, phase 3, non-inferiority trial. *Lancet Infect Dis* Published online December 20, 2023: S1473-3099(23)00650-3.

[2] Oda Y, Kumagai Y, Kanai M, et al. Comment: Persistence of immune responses of a self-amplifying RNA COVID-19 vaccine (ARCT-154) versus BNT162b2. *Lancet Infect Dis* 2024 Published online February 1, 2024. [https://doi.org/10.1016/S1473-3099\(24\)00060-4](https://doi.org/10.1016/S1473-3099(24)00060-4).

The future publications are exciting and should be in addition to already published data (see above comment).

While a previous version of the vaccine, ARCT-021, demonstrated an immune response after the first dose, this was not boosted by the second dose. In the paper currently being reviewed, the authors

used their neutralisation assay to show that there is an immune response after one dose, which is boosted by a second dose. However, it is unclear if the participants of this trial were naïve to SARS-CoV-2 at baseline. The study would benefit from more antibody data in the form of ELISA or MSD assays.

We confirm that there were significant improvements made in the development of ARCT-154 vaccine compared with ARCT-021, including the use of Spike glycoprotein of the ancestral SARS-CoV-2 strain with a D614G mutation, proline substitutions resulting in the S glycoprotein being expressed in the prefusion conformation and furin cleavage site modification to improve the stability (lines 81-2). Additional changes include optimization of replicon and modification of vaccine impurity profile. All these changes were associated with increased immunogenicity and an improved tolerability profile compared with the parent vaccine.

Please include all this information within the main body of the introduction.

We have included this information in the manuscript (lines 87-92): ARCT-154 was then developed based on the S-protein with the D614G mutation, a proline substitution resulting in the S-protein being expressed in the prefusion conformation and furin cleavage site modification to improve stability. Additional changes include optimization of the replicon and modification of the vaccine impurity profile which were associated with increased immunogenicity and an improved tolerability profile compared with the parent vaccine in preclinical studies.

We confirm that immunogenicity data presented for SARS-CoV-2 naïve population, defined as subjects without history of COVID-19 disease or COVID-19 vaccination and without anti-nucleocapsid antibodies at the time of recruitment (lines 351-2 and 427).

Please see comments in point 21, below.

Specific comments and suggestions:

1. Title: According to the CONSORT guidelines, the title should have the phrase “randomised trial” within it.

We have modified the title accordingly, to read “pooled phase 1, 2, 3a and 3b randomized, controlled trials”

Thank you

2. Reference 18 needs updating to “npj Vaccines (2022)7:161 ; <https://doi.org/10.1038/s41541-022-00590-x>”;

We have updated what is now Reference 19.

Thank you

3. Table 1 would benefit from some statistics to show that the demographics in the different phases of the trial are comparable to one another.

No statistical comparisons were made between the four independent cohorts – the results all lie

within the same range.

Noted

4. Table 2 would also benefit from some statistics to demonstrate that there is no significant difference between adverse events observed in the different phases of the trial.

A statistical comparison between the different phases would not be appropriate due to the differences in populations recruited for each phase.

Thank you for clarifying. Please can a note of this be made within the text/table so it is clear to the readers.

5. Line 122 reads “rates systemic rates”, which doesn’t make sense.

This is a typo and should read “rate of systemic adverse events” and has been corrected (lines 124-5).

Thank you for correcting this.

6. The neutralisation assay used within the study is the only immunogenicity data given within this paper. This data would be better presented and easier to interpret in graphical form rather than a table.

As already noted, the full immunogenicity data set will be presented separately so we believe a Table with geometric mean concentrations, seroconversion rates and geometric mean-fold rises in neutralizing antibodies which are considered the gold standard to measure immune responses to COVID-19 vaccines is appropriate for this paper that is primarily focused on efficacy. The neutralizing antibody response was measured using two methods which essentially confirm each other’s results.

Thank you for the clarification. I agree that neutralizing antibodies are widely considered to be the gold standard for the measurement of immune responses, which is why I believe a graphical representation of this data will be easier for the readers to interpret and therefore should be included (even if it’s in the supplementary information). As this data is a combination of phases 1-3, it is important to clearly show this information within the body of the paper, even if it will be presented (presumably in greater detail) in a full immunogenicity data paper.

We changed the presentation of immunogenicity data from Table 3 in the previous version to Figure 5 in this.

7. The neutralisation data would be further strengthened by a live neutralisation assay and the inclusion of a comparison to sera from individuals who have received a licence mRNA, as well as convalescent serum.

Again, we would emphasize that the primary objective of this paper was the efficacy in which the comparison was with placebo, not other COVID -19 vaccines. Such a comparison will be made

against licensed COVID-19 vaccines (Comirnaty and Vaxzevria) or convalescent plasma in ongoing and future papers.

I agree that the primary focus of this paper is efficacy. Therefore, I agree that the live neutralization assay may be beyond the scope of this paper. However, as the immunogenicity data has yet to be published, and as mentioned above, a graphical representation of the neutralization data already performed should be included, alongside a comparison of sera from individuals who have received a licence mRNA, as well as convalescent serum.

We would like to highlight that no comparison with licensed mRNA vaccines is possible based on results of Phase 1/2/3a/3b parts of the study. The authors intend publishing relative efficacy, immunogenicity and safety data of ARCT-154 vs AstraZeneca ChAdOx1-S vaccine in a separate manuscript, which is in preparation, based on Phase 3c of the study. As mentioned above, we have already published relative immunogenicity data vs a conventional mRNA vaccine from another clinical study in Lancet Infectious Diseases. As suggested by the reviewer, we included a graphical representation of the neutralization data from Phase 1/2/3a in the revised manuscript.

8. The authors need to be careful when using the term “seroconverting” when describing the immune response of participants to the vaccine. The recruited participants were not screened for SARS-CoV-2 S- or N- specific antibodies so it is unknown whether they have seroconverted or a low baseline immunity has been boosted by the first dose.

All study participants, included in the primary immunogenicity analysis were SARS-CoV-2 naïve, as defined by the result of anti-nucleocapsid antibody test on Day 1.

See my response to point 21 below

We are using the SCR/SRR definition proposed in the current guidelines for COVID-19 vaccine development. The term seroconvert is specifically defined in the Statistics section and indicates a four-fold or greater increase in antibody concentration from baseline to postvaccination, irrespective of the serostatus of the participant before vaccination.

Noted

9. Figure 5 - the graph begins at D35. This does not allow the reader to make a judgement as to whether efficacy can be observed after one dose and is important as licenced mRNA vaccines have demonstrated a divergence from placebo from 12 days post prime onwards. This data is only partially reported within the supplementary figures (the placebo and the vaccinees have not been separated within the supplementary table) and should be within the graph.

In Supplementary figure 2 we present the cumulative distribution curves from Day 1 that confirm the observation that the main divergence from placebo occurred after 7 days post-Dose 2

Thank you for including this figure. It makes it much clearer to the reader that the main divergence occurs 7 days post 2nd dose. It is important to note in the discussion that this is different from mRNA vaccines, where the divergence began after 1 dose.

We do not necessarily agree with the conclusion that no divergence occurred before 7 days post-Dose 2. The readers can make their own conclusion based on the presented data.

10. Figure 5 – the numbers underneath the graph do not follow what is written within the text. For example, the text states that 43 severe cases of COVID-19 were detected between days 35-92, but the figures at the bottom add up to more than 43. Were the individuals counted twice to give a cumulative score within the graph? A table to breakdown the time of COVID-19 onset would be useful (as was used in DOI: 10.1056/NEJMoa2035389; Figure 3).

The figures, as noted in the title, show COVID-19 of any severity in the upper panel, and severe COVID-19 in the lower panel. We have updated the Figure to include all the cumulative numbers of severe cases up to Day 92 showing the 43 cases (41 placebo, 2 vaccine).

It is understood that figure 5 represents COVID-19 of any severity in the upper panel, and severe COVID-19 in the lower panel. However, the confusion lay (and still lies) in where the figure of 43 comes from? If this is the total number (i.e., 41 in the placebo group and 2 in the vaccine group) then there should be an additional column in Figure 5 to denote this as I am not clear where the values come from (the total of the values in Figure 5B are 109 in the placebo group [1+3+8+13+22+29+34=109] and 7 in the vaccinee group [1+2+2+2]). I suspect these are cumulative number, but this is not explicitly clear to me.

We have updated Figure 6 (previously Figure 5) by inclusion of additional data at Day 92 with cumulative number of COVID-19 cases of any severity and severe cases in both study groups. The total number of cases on Day 92 matches those in Table 3 (previous Table 4).

11. Table 4 – have the relevant statistics been applied to justify the comment that the rates of infection were similar in male and female participants?

No pre-defined comparisons were done for subgroup analyses

Noted

12. There is no figure or table that breaks down the results of the experiments performed to record the variants responsible for the SARS-CoV-2 infections. The paper would benefit from this being graphically represented, at least in the supplementary figures.

The overwhelming majority (477 of 537 [89%]) of cases of virologically-confirmed COVID-19 in which the strain was identified were Delta. We have included this information in a new Supplementary table 5.

Thank you for including this – it is much clearer for the reader.

13. On line 209, the authors make the statement that the efficacy estimates were similar in the 1-29 day time frame, yet this period has been omitted from the results and is not clearly evident in the data presented, so this statement should be removed as it cannot be proven with the data in the current format.

We have removed this.

Thank you

14. Authors should avoid using the use of geographical location terms such as “Wuhan-Hu-1” strain. “Wildtype” is a better term.

The term Wuhan-Hu1 strain is used to indicate the genetic sequence used for manufacturing of ARCT-021 vaccine and a test strain utilized in the neutralization assay. We believe that indication of an exact strain used provides more relevant information to readers compared with ‘wildtype strain’ which actually could apply to all circulating strains of SARS-CoV-2.

I would compromise here and request the authors denote the virus as SARS-CoV-2 WT (Wuhan-Hu-1). Furthermore, if the authors prefer to use the genetic sequence identity to indicate the exact strain, then use of “Delta” should be written as Delta (B.1.617.2) as should other variants.

We updated strain nomenclature across the manuscript by adding the genetic sequence identity, as recommended.

15. On line 222, the authors use the abbreviation “VE”, which presumably represents “Vaccine Efficacy” however, this is not stated anywhere within the paper, so needs defining.

We have defined VE as vaccine efficacy (line 169).

Thank you for including this. However, the use of VE on line 222 of the original manuscript (now line 226) is the first mention of VE within the Discussion section and should, therefore, be redefined.

We have added the definition of VE in line 212.

16. The authors need to be clearer in the discussion that vaccine efficacy against the licenced mRNA vaccines was defined as “preventing COVID-19 illness, including severe disease”, rather than “severe disease and death”, which is the wider definition used used for the saRNA within this paper.

We have included “against COVID-19 illness, including severe disease,” in the text (line 218).

Thank you

17. The authors postulate the difference in efficacy is lower with their vaccine due to the timing of the efficacy trial and their definition of COVID-19 disease, but it is not clear to the reader that, when compared to mRNA vaccines, their vaccine has an efficacy of 56.6% versus 95% (BNT162b2) and 94.1% (mRNA-1273). This needs to be more explicitly stated.

First, we should indicate that this lower VE of the ARCT-154 vaccine was against Delta variant, and we have already put in text that “The effectiveness of a completed primary vaccination series of authorized COVID-19 vaccines against infection by the Delta variant has been shown to range between 46% and 91%, and from 47% to 97% against severe COVID-19 disease due to Delta variant.” (lines 222-5)

The lower VE we observed may also be a consequence of the definition of COVID-19 disease used in the trial, being based on the presence of a single symptom in combination with a positive RT-PCR. This allows the inclusion of a significant number of mild and marginally symptomatic cases in the analysis, and it is

acknowledged that the effectiveness of COVID-19 vaccination increases with the severity of COVID-19. For example, the effectiveness of a 3-dose vaccination series of BNT162b2 against asymptomatic and symptomatic COVID-19 disease during Omicron BA.2 outbreak was different - 41.4% (95% CI: 23.2–55.2; $p = 0.0001$) and 50.9% (95% CI 31.0–65.0; $p < 0.0001$), respectively. Furthermore, the effectiveness of a 3-dose series of BNT162b2 vaccine against mild, moderate and severe COVID-19, caused by the Omicron variant, within 31 to 60 days after the last vaccination, was 7.9% (95% CI: 2.3–13.1), 49.2% (95% CI: 46.8–51.4) and 76.4% (95% CI: 72.4–79.8), respectively.²² It was notable that ARCT-154 was much more efficacious against severe COVID-19 than disease of any severity. As such, the case definition used for primary efficacy analysis has a significant impact on the efficacy point estimate. Clinical studies with other vaccines used a ‘more symptomatic’ definition of COVID-19 disease (a positive RT-PCR in combination with at least two systemic symptoms or at least one the respiratory signs or symptom). (Lines 235-239)

Thank you for editing it – it is much clearer now.

The statement on lines 225-227 “it is acknowledged that the effectiveness of COVID-19 vaccination increases with the severity of COVID-19 and decreases with time since vaccination” needs referencing.

Addressed in the response above.

Thank you

18. The sentence between lines 230-234 beginning “In a population...” needs to be put into context of the vaccine being trialled. It is a valid statement, but it lacks context.

We have deleted this phrase and replaced it as above.

Thank you

19. The sentence between lines 239-241 beginning “Although the trials were not identical in design so not directly comparable...” appears to contradict itself as the authors go on to compare licenced mRNA vaccines to ARCT-154.

Indeed, we compared the results across the trials but the methodologies employed and adverse events solicited in clinical trials are sufficiently similar to allow such a general comparison.

In which case, would the authors consider rephrasing to “Although the trials were not identical in design, a general comparison determined that overall systemic AEs and local reactions were less frequent in recipients of ARCT-154 than licensed mRNA vaccines”.

We have added text to acknowledge that there may be cultural differences in reporting solicited and unsolicited reactogenicity and have added that we have now published another study in Japanese adults in whom ARCT-154 was directly compared with an mRNA vaccine (BNT162b2) and had similar reactogenicity (lines 255-258).

20. The sentence beginning on line 247 “This study probably represents...” is misleading as the authors have not tested the population used within this study for S- or N- specific antibodies to SARS-CoV-2 at baseline, so it cannot be proven that this is a truly naïve population.

Anti-N seropositivity was tested at baseline in all participants and the primary efficacy analysis and presented immunogenicity analyses only include SARS-CoV-2 naïve subjects based on medical history and serology for N-

specific antibodies. This is now indicated (lines 337-8 and 408).

Thank you for including this within the text – however, I would ideally like to see this in supplementary results and then the results referenced within the text. Having trawled through all the datasets, I cannot see any evidence that testing for N-antibodies was performed. Indeed, within the “exclusion criteria” in supplementary table 2, point 3, line 613 reads “nucleocapsid positive test is not exclusionary”, which seems to contradict the statements within the body of the paper.

Testing for N-protein is mentioned in the Methods (lines 351-2), and we have included the N-protein serology results in Table 1 showing less than 1% of participants on phase3 b were positive.

21. The authors need to comment that licenced mRNA vaccines are/have been updated to reflect the changes in circulating strains. These updated vaccines are being given to older/at risk populations in many countries, which is helping to boost the waning immunity within these populations. saRNA vaccines can, of course, be an additional tool.

We have added a comment to this effect (lines 283-287).

Thank you

22. The authors need to reference Szubert et al (2023) in line 258 who also demonstrated that a SARS-CoV-2 saRNA vaccine was capable of boosting antibodies and neutralisation activity in individuals with pre-existing immunity.

We were unaware of this manuscript at the time of writing but are grateful to the reviewer for pointing it out and have now added it to our manuscript (lines 272-275).

Thank you

Reviewer #4 (Remarks to the Author):

Per the Editor's request, this reviewer focused rather narrowly on determining if Reviewer #2's comments were adequately addressed in the revision. Comments which were considered adequately resolved are not repeated in this review, and only items remaining to be addressed are highlighted here:

1. Reviewer #2 found it difficult to match numbers vaccinated across Tables 1, 2, 4, and the CONSORT diagram. The rebuttal cites changes in terminology and minor corrections of numbers in Figure 1 (actually, Figure 2). There is at least one remaining issue: the column N's used in Table 2 cannot be found on the CONSORT, and it is unclear why. Per the protocol, the Safety Set should have been used for Table 2, defined as the number vaccinated. The numbers vaccinated in Phases 1/2/3a/3b in the CONSORT differ from column N's in table 2. In addition, Table 1 would appear to be a summary among the Randomized Set with n=750 randomized to receive ARCT-154 in phase 1/2/3a (i.e., there is no "Enrolled Set" defined in the protocol), but Figure 2 shows 749 randomized into the ARCT-154 arm in phase 1/2/3a; I did not identify a reason for these minor differences. The analysis population, using terminology from the protocol, should be identified in the caption or footnotes for all tables.

We sympathize with the reviewer's concerns as evidently this was a very complicated study and small variations in numbers, e.g. due to randomized participants receiving the wrong study treatment (placebo instead of vaccine, or vice versa, or no treatment) make it difficult to follow. We have attempted to assist in this by adding footnotes to the relevant tables to explain where such minor issues occurred allowing the reader to reconcile the numbers

2. Reviewer #2 did not find sufficient evidence to warrant the claim that solicited AEs were lower in the present study relative to those from external studies of alternative mRNA COVID-19 vaccines. The revised manuscript aims to strengthen the evidence supporting the claim, citing similarity of data collection/reporting methods. However, Reviewer #2's main concern has not been fully addressed, as a number of issues should be either accounted for in a supportive analysis to strengthen the claim, or instead cited as weaknesses of the comparison including potential cultural differences in reporting of subjective events such as headache and myalgia, as well as potentially varying mixtures of ages included in each study, which is generally understood to be related to rates of reactogenicity for COVID-19 vaccines.

We admit that there are variations in reporting of solicited and unsolicited adverse events between populations, but this study was controlled with a placebo. Further, similar "reactogenicity" has now been reported in Japanese adults in a published paper which was submitted after the present manuscript in which there was a head-to-head comparison with an mRNA vaccine [Oda et al].

Overall the manuscript appeared well-written, presenting noteworthy efficacy results for the novel self-amplifying mRNA vaccine candidate. A suggestion to the authors for the analysis following Day 92 is to take great care with how to present efficacy data that will presumably have two different risk periods for vaccinees (full duration for those initially vaccinated and post-D92 for controls who are switched over, e.g. no controls in the post-D92 period). Similar considerations will apply for safety (i.e., lumping together reactions to placebo doses may or may not be appropriate for those initially given placebo injections versus those given placebo injections at D92, after having previously received two vaccinations). This seems partially, but not fully addressed in the SAP shared for the present analysis.

We appreciate that the reviewer has read the full study protocol, but would note that that protocol includes additional phases of this study which we have not attempted to report – notably the switch-over at Day 92 when placebo recipients received the vaccine and vaccinees received placebo to ensure all participants could benefit from immunization. There was also a phase 3c study with a direct comparison of ARCT-154 with the adenovirus-vector vaccine, ChAdOx1-S. As already noted, this is a very large study and manuscript and so we only provide data up the switch-over at Day 92. These additional aspects of this study will be reported separately in switch-over and phase 3c manuscripts which are currently in preparation.

To clarify this point we have added text about the switch-over at Day 92 (lines 354-359).